# Improved State Mixing in Higher-Order and Block Diagonal Linear Recurrent Networks

## Abstract

Linear recurrent networks (LRNNs) and linear state space models (SSMs) promise computational and memory efficiency on long-sequence modeling tasks, yet their diagonal state transitions limit expressivity. Dense and/or nonlinear architectures (e.g., LSTMs) on the other hand are provably more expressive, but computationally costly. Here, we explore how expressivity in LRNNs can be increased via richer state mixing across time and channels while maintaining competitive efficiency. Specifically, we introduce two structured LRNN architectures: (i) Higher-order Linear Recurrent Units (H-LRU), which generalize first-order recurrence to $m$-th order, mixing multiple past states, and (ii) Block-Diagonal LRUs (BD-LRU), which enable dense intra-block channel mixing. Per-channel (H-LRU) / per-row (BD-LRU) L1-normalization of selective gates stabilizes training and allows for scaling window/block sizes. In synthetic sequence-modeling benchmarks (compression, selective copying, associative recall), H-LRU is found to be the most parameter-efficient in compression, while the performance of BD-LRU matches or exceeds those of linear SSMs (Mamba), low-rank LRNNs (DeltaNet) and LSTM baselines. In permutation composition tasks ($S_3$-$S_5$), BD-LRU is found to efficiently solve these tasks at moderate block sizes, outperforming both linear and non-linear baselines. A parallel-scan implementation of the proposed architectures keeps the throughput competitive with diagonal LRNNs for moderate orders (H-LRU) and block sizes (BD-LRU), while preserving the efficiency that motivated LRNNs. These results indicate that the structure of state mixing rather than width alone shapes expressivity of LRNNs, offering a practical route to closing the efficiency–expressivity gap in linear sequence models.

## 1 Introduction

Recent studies have highlighted fundamental limitations of linear recurrent networks (LRNNs) by showing that the structure of the state-transition matrix results in a trade-off between efficiency and expressivity (Merrill and Sabharwal, 2023; Cirone et al., 2024; Merrill et al., 2024). Architectures based on diagonal matrices enable an efficient implementation but are inherently limited in expressive power, while dense models are provably more expressive yet computationally prohibitive. To bridge this gap, several LRNN architectures have been proposed: efficient structured architectures such as ones with diagonal-plus-low-rank matrices (Yang et al., 2024a; Peng et al., 2025) and their products (Siems et al., 2025), ones based on approximations of dense matrices at test time (Sun et al., 2024; Movahedi et al., 2025; von Oswald et al., 2025), and other solutions that are *de facto* equivalent to block-diagonal architectures (e.g., oscillatory blocks (Rusch and Rus, 2024) and complex-valued states (Orvieto et al., 2023; De et al., 2024)). Together, these studies suggest that exploring the configuration space between diagonal and dense transition matrices may yield more expressive LRNN models.

When designing block-diagonal recurrences, the immediate issue one faces is that of dynamical stability and forward pass normalization – a crucial element that is well studied and discussed in diagonal LRNNs (Orvieto et al., 2023; Wang and Li, 2023; Zucchet and Orvieto, 2024), yet requires additional care in non-diagonal linear architectures where eigenvalues are not readily available. Traditionally, stability has been ensured by parameterizations that constrain eigenvalues of the transition matrix inside the complex unit disk (Arjovsky et al., 2016; Helfrich et al., 2018), a strategy that effectively mitigates vanishing and exploding gradients. More recently, similar conditions have

been applied to derive efficient reparameterizations that ensure stability in diagonal linear recurrent units (Orvieto et al., 2023; De et al., 2024). In both selective and non-selective SSMs (designed in continuous-time), stability is achieved by exponential parametrization, resulting from zero-order-hold discretization techniques (Gu et al., 2021; Gu and Dao, 2023). Finally, in LRNNs with diagonal-plus-low-rank transition matrices, normalization arises naturally from the structure of generalized Householder transformations (Yang et al., 2024b). Although several recent studies have examined block-diagonal architectures, they either focus on parameterizations of non-selective models (Biegun et al., 2024; Rusch and Rus, 2024; Walker et al., 2025), analyze only the stability of the state-transition matrix norm (Fan et al., 2023), or rely on architectures where this matrix is normalized by design (Yang et al., 2024b), without fully addressing the problem of joint normalization of selective state-transition matrix and selective input gates, which has been previously shown critical for sequence modeling in diagonal LRNNs Orvieto et al. (2023); Gu and Dao (2023); De et al. (2024).

Building on this line of work, we explore how to improve expressivity of LRNNs through structured selective state mixing, while preserving their computational efficiency. Starting from basic consider-ations, we introduce two architectures with such mixing: (i) Higher-order Linear Recurrent Units (H-LRU), which generalize first-order recurrence to $m$-th order, which allow for mixing multiple past states, and (ii) Block-Diagonal LRUs (BD-LRU), which enable dense intra-block channel mixing. We equip these models with input-dependent selective gates which are restricted by per-channel/row L1 normalization. This normalization allows both architectures to effectively scale with window or block size, respectively, and achieve competitive or superior accuracy to diagonal, low-rank and non-linear baselines on a set of synthetic sequence modeling tasks. In addition, a parallel-scan implementation maintains high throughput for moderate block sizes, preserving the efficiency that motivates linear recurrences. Overall, contrary to the common belief that width alone determines performance, our results indicate that expressivity is primarily shaped by the structure of state mixing.

## 2 Higher-order and block diagonal linear recurrent networks.

Modern linear recurrent models (e.g., S4, LRU, Mamba), as well as linear attention models (e.g. GLA, DeltaNet), exchange information between tokens by means of a recurrent mechanism

$$\mathbf{h}_t = \mathbf{a}_t \odot \mathbf{h}_{t-1} + \mathbf{b}_t \odot \mathbf{v}_t, \tag{1}$$

where $\mathbf{h}_t \in \mathbb{R}^N$ is the hidden state computed at time $t$, and $\mathbf{a}_t, \mathbf{b}_t$ are input-dependent and potentially state-dependent gates prescribing how current information $\mathbf{v}_t = \mathbf{W}_\mathbf{v}\mathbf{x_t}$ (pointwise function of the input $\mathbf{x}_t$) gets stored in the network state.

Through this mechanism the output of the network at time $t$, a function of the hidden state $\mathbf{h}_t$, can access information about past inputs $\mathbf{v}_1, \mathbf{v}_2, \ldots, \mathbf{v}_t$. In fact, one can write in closed form $\mathbf{h}_t = \sum_{i=1}^t (\prod_{j=t-i}^t \mathbf{a}_j) \odot \mathbf{b}_i \odot \mathbf{v}_i$. However, as is well known from both modern and classical literature, the system above suffers from vanishing gradients with respect to the inputs (Pascanu et al., 2013; Wang and Li, 2023; Zucchet and Orvieto, 2024). Standard approaches to address this issue are to re-parametrize the entries of $\mathbf{a}_t$ such that they absolute values are close to a value of $1$ (Orvieto et al., 2023), and to increase the dimensionality of $\mathbf{h}_t$ (Orvieto et al., 2024). Although it can be shown that this strategy can help memorization (Arora et al., 2023; Okpekpe and Orvieto, 2025), it is also known that going beyond diagonal formulations – i.e. mixing the hidden state as $\mathbf{A}_t\mathbf{h}_{t-1}$ instead of $\mathbf{a}_t \odot \mathbf{h}_{t-1} = \mathrm{diag}(\mathbf{a}_t)\mathbf{h}_{t-1}$ – can drastically improve performance on challenging reasoning tasks involving state-tracking (Merrill et al., 2024; Cirone et al., 2024; Movahedi et al., 2025).

An *orthogonal* approach to diagonal state expansion that we consider here, is to instead design recursions of *higher complexity*. An example in recent literature comes from (Rusch and Rus, 2024), where the authors consider system equations given by the second-order oscillatory ordinary differential equation $\mathbf{h}''(t) = -\bar{\mathbf{a}}(t) \odot \mathbf{h}(t) + \bar{\mathbf{b}}(t) \odot \mathbf{v}(t)$. After discretization[1], this leads to a second-order difference equation of the form

$$\mathbf{h}_t = \mathbf{a}_{1,t} \odot \mathbf{h}_{t-1} + \mathbf{a}_{2,t} \odot \mathbf{h}_{t-2} + \mathbf{a}_{0,t} \odot \mathbf{v}_t, \tag{2}$$

where coefficients $\mathbf{a}_{i,t}$ are a function of the discretization method. Notably, the model 2 can already be made more expressive if we allow arbitrary selective gates $\mathbf{a}_{1,t}, \mathbf{a}_{2,t}, \mathbf{a}_{0,t}$ in contrast to the fixed parameterization of discretization schemes.

---

[1]Plugging in the second-order backward estimate $\mathbf{h}''(t)\Delta \simeq \mathbf{h}_t - 2\mathbf{h}_{t-1} + \mathbf{h}_{t-2}$ (Hairer et al., 1993).

**Higher-order Recurrence**   Inspired by Eq. 2, we generalize Eq. 1 and introduce Higher-order Linear Recurrent Units (H-LRUs) as follows:

$$\mathbf{h}_t = \sum_{i=1}^{m} \mathbf{a}_{i,t} \odot \mathbf{h}_{t-i} + \mathbf{a}_{0,t} \odot \mathbf{v}_t. \tag{H-LRU}$$

This parametrizes the state evolution by an $m$-th order difference equation. Such models are a standard tool in time series statistics for forecasting (ARMA processes, see e.g. Hamilton (2020)) and are *canonical* in systems theory, since they lead to minimal realization (i.e., with provably the smallest memory size) of linear dynamical systems (Glad and Ljung, 2018).

To see the connection with controllable canonical forms for transition matrices in systems theory, it is sufficient to denote by $h_{t-1}^k$ the $k$-th coordinate ($k \in \{1, 2, \dots, N\}$) of $\mathbf{h}_t$ and by $a_{i,t}^k$ the $k$-th coordinate of $\mathbf{a}_{i,t}$. Then, with $\times$ denoting the standard matrix multiplication,

$$\mathbf{h}_t^k = \mathbf{A}_t^k \times \mathbf{h}_{t-1}^k + \mathbf{a}_{0,t}^k \odot \mathbf{v}_t^k,$$

$$\mathbf{A}_t^k = \begin{bmatrix} a_{1,t}^k & \cdots & a_{m-1,t}^k & a_{m,t}^k \\ 1 & \cdots & 0 & 0 \\ \vdots & \ddots & \vdots & \vdots \\ 0 & \cdots & 1 & 0 \end{bmatrix}, \ \mathbf{h}_{t-1}^k = \begin{bmatrix} h_{t-1}^k \\ \vdots \\ h_{t-m}^k \end{bmatrix}, \ \mathbf{a}_{0,t}^k = \begin{bmatrix} a_{0,t}^k \\ \vdots \\ 0 \end{bmatrix}, \ \mathbf{v}_t^k = \begin{bmatrix} v_t^k \\ \vdots \\ 0 \end{bmatrix}, \tag{3}$$

where $\mathbf{A}^k$ is a structured companion-like matrix which allows richer dynamic modes (e.g. oscillatory modes). Though eigenvalues for $\mathbf{A}_t^k$ are not available in closed form[2], dynamical stability for the system above can be guaranteed and is crucial for performance, as we will discuss in the next section.

**Block Diagonal Representation.**   The substitution in Eq. 3 allows us to rewrite the system equations in H-LRU as a generalized first-order recurrence

$$\mathbf{h}_t = \mathbf{A}_t \times \mathbf{h}_{t-1} + \mathbf{a}_{0,t} \odot \mathbf{v}_t, \tag{4}$$

$$\mathbf{A} = \mathrm{diag}(\mathbf{A}_t^1, \dots, \mathbf{A}_t^N), \ \mathbf{h}_{t-1} = \begin{bmatrix} \mathbf{h}_{t-1}^1 \\ \vdots \\ \mathbf{h}_{t-1}^N \end{bmatrix}, \ \mathbf{a}_{0,t} = \begin{bmatrix} \mathbf{a}_{0,t}^1 \\ \vdots \\ \mathbf{a}_{0,t}^N \end{bmatrix}, \ \mathbf{v}_t = \begin{bmatrix} \mathbf{v}_t^1 \\ \vdots \\ \mathbf{v}_t^N \end{bmatrix},$$

revealing that the H-LRU architecture corresponds to a recurrent network with a structured block diagonal state-transition matrix.

Independently, we also investigate a second kind of recurrence with complexity higher than the diagonal case, the block diagonal linear recurrent unit (BD-LRU). In contrast to the structured temporal state mixing implemented inside H-LRU blocks, BD-LRU implements dense channel mixing inside all blocks for all vectors and matrices by setting

$$\mathbf{h}_t^k = \mathbf{A}^k \times \mathbf{h}_{t-1}^k + \mathbf{a}_{0,t}^k \odot \mathbf{v}_t^k, \tag{BD-LRU}$$

$$\mathbf{A}_t^k = \begin{bmatrix} a_{1,1,t}^k & \cdots & a_{1,m-1,t}^k & a_{1,m,t}^k \\ a_{2,1,t}^k & \cdots & a_{2,m-1,t}^k & a_{2,m,t}^k \\ \vdots & \ddots & \vdots & \vdots \\ a_{m,1,t}^k & \cdots & a_{m,m-1,t}^k & a_{m,m,t}^k \end{bmatrix}, \ \mathbf{h}_{t-1}^k = \begin{bmatrix} h_{1,t-1}^k \\ \vdots \\ h_{m,t-1}^k \end{bmatrix}, \ \mathbf{a}_{0,t}^k = \begin{bmatrix} a_{1,0,t}^k \\ \vdots \\ a_{m,0,t}^k \end{bmatrix}, \ \mathbf{v}_t^k = \begin{bmatrix} v_{1,t}^k \\ \vdots \\ v_{m,t}^k \end{bmatrix}. \tag{5}$$

As for H-LRU (Eq. 4), the block size $m$ of BD-LRU corresponds to the size of a square matrix $\mathbf{A}^k$ and $k \in [1, N]$ corresponds to the block index of this matrix. The hidden size of BD-LRU is equal to the extended block diagonal representation of the H-LRU architecture. But in contrast to H-LRU (Eq. 4), all vectors $\mathbf{a}_0^k, \mathbf{h}_t^k, \mathbf{v}_t^k \in \mathbb{R}^m$ and all matrices $\mathbf{A}^k \in \mathbb{R}^{m \times m}$ in BD-LRU are dense and there is no dependence on the several previous hidden states that is characteristic of the H-LRU architecture. Importantly, the structure of BD-LRU does not allow for the same eigenvalue analysis as is possible for H-LRU. Yet, as we show in the next section, we can guarantee its dynamical stability using a normalization technique similar to that of H-LRU.

To endow the models with input selectivity, we introduce input-dependent gates for both H-LRU ($a'_{j,t} = \textit{Linear}_j(\mathbf{x}_t)$) and BD-LRU ($a'_{i,j,t} = \textit{Linear}_{i,j}(\mathbf{x}_t)$). Fig. 1 provides a schematic illustration of the proposed gating mechanisms in block-diagonal form, showing both the state gates that form the state-transition matrix and the input gates applied to external inputs.

---

[2] Solve the equation $\chi_{\mathbf{A}^k}(\lambda) = \det(\lambda I - \mathbf{A}^k) = \lambda^m - a_{1,t}^k \lambda^{m-1} - a_{2,t}^k \lambda^{m-2} - \cdots - a_{m-1,t}^k \lambda - a_{m,t}^k = 0$.

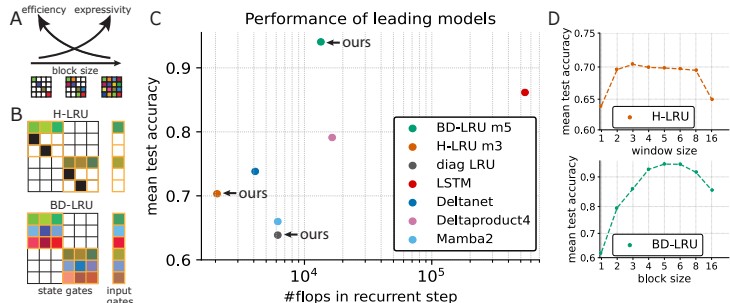

Figure 1: Structure and performance of the proposed H-LRU and BD-LRU architectures. **A.** A schematic illustration of the theoretically predicted trade-off between expressivity and efficiency of block-diagonal linear recurrent networks. **B.** Schematic illustration of the gating mechanisms in block-diagonal form, showing both the state gates that constitute the state-transition matrix and the input gates that act on external inputs. The structure of the gates' selectivity is color-coded: white squares indicate fixed zero gates, black squares indicate fixed identity gates, other colors indicate active selective gates; similar color palettes indicates row-wise normalization. **C.** Summary of the performance of the proposed and the baseline models. The *x*-axis indicates the number of FLOPs per recurrent step. The *y*-axis denotes the mean test accuracy over all considered tasks (compression, selective copying, in context recall, permutation) of the overall best performing model configuration (hidden size up to 6k). Optimal hidden sizes vary between models, see also Fig. 5. Note that H-LRU and BD-LRU can achieve better or matching performance than both linear and non-linear baselines while requiring fewer FLOPs per recurrent step. Diagonal LRU presents the best results across both H-LRU m1 and BD-LRU m1, which are identical models for $m = 1$. **D.** Best performance for different window sizes $m$ (H-LRU) and block sizes $m$ (BD-LRU).

## 3  NORMALIZATION

Normalization schemes for RNNs which impose restrictions on the eigenvalues of the state-transition matrix have proven to be very effective as they directly address the vanishing and exploding gradient problem (Pascanu et al., 2013). This approach has led to the development of a variety of models with restrictions on the norm of the state-transition matrix (Arjovsky et al., 2016; Helfrich et al., 2018). More recently, similar normalization techniques were applied to exponentiated gates in linear recurrent units (LRU, Orvieto et al. (2023)) and optimized discretization schemes in state space models (SSM, Gu et al. (2021)). However, as detailed in Orvieto et al. (2023), stability in a dynamical systems sense (i.e., requiring that the eigenvalues of the hidden-to-hidden transition be less than one in absolute value) does not necessarily guarantee a properly normalized forward pass in this case. This can negatively affect performance, as discussed in the next section.

To understand this phenomenon, one can consider the trivial one-dimensional linear setting $h_t = a h_{t-1} + b x_t$, where $x_t = 1$ for all $t$. For $a \in (0,1)$, as $t \to \infty$, $h_t$ converges to the value $(1-a)^{-1}b$, which can be substantially greater than 1 if $a$ gets close to 1, as allowed and incentivized by recent sigmoidal parametrizations (Orvieto et al., 2023). Of course, the forward-pass norm in this case is preserved if input and forget gates are adapted, that is, if we consider RNNs of the form $h_t = a h_{t-1} + (1-a)x_t$, i.e., $b = 1-a$. This directly translates to the case of a diagonal network where models such as S4 (Gu et al., 2020) and Mamba (Gu and Dao, 2023) adopt a forget gate of the form $a = e^\Delta$, coupled with an input gate $b = \Delta \approx (1-a)$ if $\Delta$ is close to zero. As suggested also directly from the original GRU formulation (Cho et al., 2014) as well as recent works (Feng et al., 2024), for the diagonal setting (coinciding with $m = 1$ in H-LRU and BD-LRU) it is convenient to start by adapting Eq. 1 to $\mathbf{h}_t = \mathbf{a}_t \odot \mathbf{h}_{t-1} + (1 - \mathbf{a}_t) \odot \mathbf{v}_t$. Stability for $m \geq 1$ is guaranteed when choosing coefficients as prescribed by the next proposition.

**Proposition 1** *Consider either the H-LRU or the BD-LRU architectures, written in matrix form as shown in Equations 3 and 5. If for any $k \in [1, N]$, the k-th recurrent non-diagonal block $\mathbf{h}_t^k = \mathbf{A}_t^k \times \mathbf{h}_{t-1}^k + \mathbf{a}_{0,t}^k \odot \mathbf{v}_t^k$ is such that the matrix $\mathcal{A}_t^k := [\mathbf{A}_t^k, \mathbf{a}_{0,t}^k] \in \mathbb{R}^{m \times (m+1)}$ has the property*

that $\sum_{j=1}^{m+1} |(\mathcal{A}_t^k)_{i,j}| = 1$ *for every row* $i \in [1, m]$*, then the recurrence is stable from a dynamical* *systems perspective and the forward pass is normalized, meaning that* $\|\mathbf{h}_T\|_\infty \leq \max_{t \in [0,T]} \|\mathbf{v}_t\|_\infty$.

The proposition above suggests that to achieve a normalized forward pass, L1-normalization should be applied to raw selective gates. For H-LRU, it is sufficient to normalize over all $m + 1$ coefficients of the $m$-th order recurrence, while for BD-LRU, we apply a row-wise normalization over the hidden state gates and the input gate. Let us therefore denote as $a's$ the raw gates (linear functions of the input) before normalization. We set

$$\textbf{H-LRU: } a_{j,t} = \frac{f(a'_{j,t})}{\sum_{l=0}^{m} f(a'_{l,t})}; \quad \textbf{BD-LRU: } a_{i,j,t} = \frac{f(a'_{i,j,t})}{\sum_{l=0}^{m} f(a'_{i,l,t})}, \tag{6}$$

where $f(\cdot)$ is a gate parametrization function; the block index is omitted for clarity. Note that this normalization only affects the elements inside on-diagonal blocks and has no impact on off-diagonal blocks (consisting of zero matrices). Note that the introduced normalization restricts eigenvalues of the state-transition matrix to be smaller than the $L1$ norm of the corresponding row, meaning that the eigenvalues of the state-transition matrix are limited by a value of the input gate

$$|\lambda_{i,t}| \leq \sum_{l=1}^{m} |a_{i,l,t}| = 1 - |a_{i,0,t}|, \tag{7}$$

where $i$ is the channel index in H-LRU or row index in BD-LRU. This results in a joint normalization for input and state gates that allows selective block-diagonal LRNNs to balance attention to hidden states and inputs in a similar way as in first-order non-selective and selective LRUs (Orvieto et al., 2023; De et al., 2024). This is in contrast to previous studies on selective block-diagonal LRNNs that only addressed the stability of the state-transition matrix (Fan et al., 2023).

Although the introduced normalization guarantees the stability of the recurrence, it has been shown that gradient-based learning is also highly sensitive to the specific choice of parametrization (Zucchet and Orvieto, 2024). In contrast to the normalization used in non-selective block-diagonal LRNNs that rely on structured parameterizations such as discretization schemes (Rusch and Rus, 2024; Walker et al., 2025), joint parametrization of the state-transition matrices and input gate (Biegun et al., 2024), and exponential reparametrization (Orvieto et al., 2023), our proposed normalization is more general as it can be applied to variety of both non-selective and selective parameterizations. This allowed us to independently evaluate several variants of gate parametrizations that are defined by the function $f$ in Eq. 6. As can be seen in Fig. 2, *our normalization strategy greatly improves performance* of both H-LRUs and BD-LRUs.

## 4 EXPERIMENTS ON TOKEN MANIPULATION TASKS

The sequence modeling capabilities of modern neural architectures are often evaluated through large-scale experiments involving models with billions of parameters and trained on trillions of tokens (Kaplan et al., 2020; Waleffe et al., 2024). However, recent studies have shown that many of these capabilities can be assessed using smaller models trained on carefully designed synthetic datasets which target specific tasks that are crucial for general sequence modeling (Arora et al., 2023; Poli et al., 2024).

First, the well-established equivalence between lossless compression and probabilistic modeling suggests that models that compress well also generalize well (Shannon, 1948; Hutter, 2005). Indeed, recent work shows that there is a clear connection between language modeling and compression (Gu, 2025), although with some difference in scaling laws (Delétang et al., 2023). In light of this, we include in our evaluation a task that tests the efficiency of temporal information integration, the auto-encoding compression task from Poli et al. (2024).

Next, general sequence modeling requires not only the ability to develop a fixed prediction algorithm, but also the capacity to adapt dynamically to changes within the input context. Such *in-context abilities* have been extensively studied and have been suggested to explain the success of the Transformer architecture (Olsson et al., 2022). To benchmark this basic capability, we choose the selective copying and associative recall tasks that have been shown to be good indicators of the in-context abilities of sequence models (Arora et al., 2023; Poli et al., 2024), as well as indicators of downstream capabilities (Waleffe et al., 2024).

**Normalization allows scaling with window size.** The specifics of parametrization play a crucial role in the sensitivity of parameters under gradient-based learning, especially in the context of RNNs (Zucchet and Orvieto, 2024). In Section 3, we derived a parametrization/normalization strategy on input and forget gates that guarantees forward pass stability, following insights from previous literature (Orvieto et al., 2023). Here, we show that our normalization strategies are crucial for performance. We tested several variants of the function $f$ for L1 normalization in 6: exponentiated gate $exp(\cdot)$ (softmax normalization), sigmoidal gates $\sigma(\cdot)$, ReLU gates $relu(\cdot)$. As a baseline, we also tested all models without normalization.

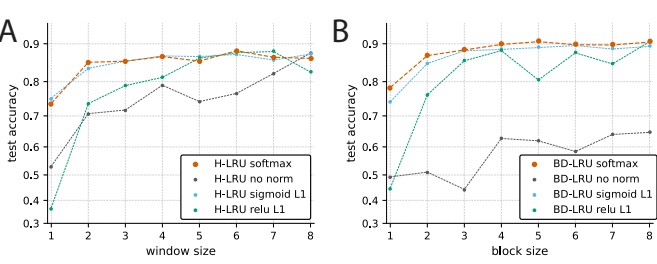

Figure 2: Scaling of performance with window/block size on the compression task for L1 normalization with different parameterizations. Results are shown for different window/block sizes $m$ of the higher-order LRU (H-LRU) and block diagonal LRU (BD-LRU). **A**. Comparison between H-LRUs. **B**. Comparison between BD-LRUs.

We found that both softmax and sigmoidal L1 normalizations allowed the models to effectively scale with window and block size, see Fig. 2. Without normalization and with the ReLU normalization, both H-LRU and BD-LRU improve at lower rate with window size. With softmax or sigmoidal L1 normalizations, the improvement with window size was especially pronounced between a window/block size of 1 and 2. Our eigenvalue analysis (see Appendix J) indicates that this gain corresponds to the emergence of negative eigenvalues, consistent with the findings of Grazzi et al. (2024). We also observe that further improvements in performance are associated with a broader range of complex eigenvalues, which are enabled starting from the block size 3. These results also align well with previous studies on beneficial role of oscillatory dynamics in recurrent networks (Rusch and Mishra, 2021; Effenberger et al., 2022; Dubinin and Effenberger, 2024; Rusch and Rus, 2024).

We noticed that for moderate block sizes ($m \in [2, 5]$), the softmax normalization performed comparable or better than sigmoidal normalization, making this the default choice for all the remaining experiments. That also agrees with previous findings that exponentiation of the gates benefit gradient-based learning (Orvieto et al., 2023; Zhang et al., 2024).

**Scaling with hidden state is limited by state mixing.** Next, we performed experiments in which we investigated the difference between scaling the window size and the hidden size. In these experiments we found that for both H-LRUs and BD-LRUs, the scaling with hidden size could not compensate for a lack of expressivity. In other words, window/block size was found to be the key factor for performance, see Fig. 5. We also found that scaling of H-LRUs and BD-LRUs results in models that are competitive with LSTMs and achieve higher performance than other linear recurrent baselines, both diagonal ones such as Mamba and low-rank ones such as DeltaNet and DeltaProduct, see Table 1. In line with the observed limitations of diagonal RNNs, we found that scaling the hidden size in a Mamba model also had limited effect on performance, see Fig. 5. Notably, we also found distinct scaling behaviors for the compression and our other tasks, aligning with previous results Delétang et al. (2023). In the compression (auto-encoding) task, models with smaller block size outperformed larger counterparts, while performance on autoregressive tasks scaled positively with block size. Therefore, the decrease in aggregate performance for larger block sizes is substantially driven by the results on the compression task.

Our scaling experiments show a direct trade-off between parameter efficiency and peak performance, as governed by the block and window sizes for BD-LRU and H-LRU, respectively. Models with smaller block/window sizes saturate in performance at lower parameter counts, demonstrating high efficiency. In contrast, models with larger block/window sizes require a larger hidden dimension to match the performance of the smaller models, but can ultimately achieve a much higher performance. This indicates that richer state mixing increases a model's expressive power at the expense of parameter efficiency.

**H-LRUs are parameter efficient.** We also found that in the compression task which does not require complex token manipulation, H-LRU demonstrated the most parameter efficient scaling with

| Models | Recall | Copy | Compress | Overall |
|---|---|---|---|---|
| LSTM | **1.000** | **1.000** | 0.750 | 0.916 |
| Mamba2 | **1.000** | 0.807 | 0.720 | 0.842 |
| Deltanet[-1,1] | **1.000** | 0.892 | 0.782 | 0.892 |
| Deltaproduct$_4$[-1,1] | **1.000** | **1.000** | 0.717 | 0.906 |
| BD-LRU m1 (ours) | 0.775 | 0.835 | 0.725 | 0.778 |
| BD-LRU m2 | **1.000** | 0.962 | 0.760 | 0.908 |
| BD-LRU m3 | **1.000** | 0.980 | 0.762 | 0.916 |
| BD-LRU m5 | **1.000** | 0.985 | 0.782 | **0.922** |
| BD-LRU m8 | **1.000** | 0.992 | 0.748 | 0.913 |
| H-LRU m1 (ours) | 0.785 | 0.848 | 0.760 | 0.797 |
| H-LRU m2 | 0.998 | 0.855 | 0.770 | 0.874 |
| H-LRU m3 | **1.000** | 0.855 | 0.772 | 0.876 |
| H-LRU m5 | **1.000** | 0.838 | 0.775 | 0.871 |
| H-LRU m8 | **1.000** | 0.810 | 0.768 | 0.859 |

Table 1: Performance on the in-context recall, selective copying and compression tasks. The presented results are the average of best test accuracies across four configurations of the corresponding synthetic dataset with different vocabulary sizes, sequence lengths and number of training examples. Results are shown for different window (H-LRU) abd block sizes (BD-LRU) $m$. Note that overall performance of our models consistently improves with window/block size up to approximately 3–5, after which the gains saturate or exhibit slight degradation. All models are single-layer configurations with a maximum overall hidden dimension of 6144. See Appendix C for extended table.

hidden size, achieving accuracies not accessible to Mamba and LSTM of similar sizes (in terms of the number of trainable parameters), see Fig. 5. This aligns well with our predictions that the inductive bias introduced by extended temporal mixing results in hidden representations with better compression capabilities.

**BD-LRUs are expressive across tasks.** In contrast to the compression task, the selective copying task requires more extensive token manipulation. We found that the performance of BD-LRUs scales more favorably with hidden size than the one of H-LRUs. Furthermore, BD-LRUs were able to outperform Mamba and DeltaNet, achieving performance that is competitive with LSTMs and DeltaProduct. At the same time, BD-LRUs achieved the best performance also in the compression task. Overall, the introduced normalization scheme allows BD-LRU to efficiently utilize the expressivity of their dense block diagonal structure to approximate a variety of mixing patterns and to achieve the best overall results on our set of synthetic tasks, see Table 1.

## 5 EXPERIMENTS ON PERMUTATION TASKS

An important property of dense recurrent networks is that one layer of such model can easily solve inherently sequential tasks such as permutation composition. In theory, linear diagonal networks and Transformers can also solve any of these tasks, but only if we assume an infinite depth approximation. In practice, it has been shown that they cannot effectively approximate the evolution of recurrent state with a bounded number of layers (Merrill et al., 2024). Furthermore, it was proposed that there is a parallelism-expressivity trade-off, in which efficient parallelization comes at the expense of decreased expressivity (Merrill and Sabharwal, 2023).

To evaluate the ability of a model to learn a permutation structure from data, we use a synthetic dataset based on the symmetric group $S_n$ - the group of all permutations over $n$ elements (Merrill et al., 2024). Each instance in the dataset corresponds to a specific permutation sampled from $S_n$, and the model is tasked with learning the mapping that defines the permutation purely from input-output examples within a sequence. We evaluated model performance on a series of increasingly complex permutation learning tasks derived from the symmetric groups $S_2$ through $S_5$.

**BD-LRUs efficiently learn permutations.** All tested recurrent architectures (H-LRU, BD-LRU, LSTM, Deltanet, Deltaproduct) were able to perfectly solve the $S_2$ task, which represents a uniquely simple permutation group as it is also a commutative cyclic group. However, as the group order increases over $S_3$ to $S_5$, the non-commutative structure of the permutation tasks increasingly posed

| Models | $S_3$ (10k samples) | $S_3$ (250) | $S_4$ (50k) | $S_4$ (3k) | $S_5$ (100k) | Overall |
|---|---|---|---|---|---|---|
| LSTM | **1.000** | 0.320 | **1.000** | 0.370 | **1.000** | 0.738 |
| Mamba2 | 0.660 | 0.280 | 0.430 | 0.120 | 0.260 | 0.350 |
| Deltanet[-1,1] | **1.000** | 0.260 | 0.470 | 0.140 | 0.140 | 0.402 |
| Deltaproduct$_4$[-1,1] | **1.000** | 0.270 | **1.000** | 0.130 | 0.140 | 0.508 |
| BD-LRU m1 (ours) | 0.560 | 0.380 | 0.340 | 0.220 | 0.210 | 0.340 |
| BD-LRU m2 | **1.000** | 0.490 | 0.700 | 0.360 | 0.340 | 0.576 |
| BD-LRU m3 | **1.000** | **1.000** | **1.000** | 0.430 | 0.480 | 0.782 |
| BD-LRU m5 | **1.000** | **1.000** | **1.000** | **1.000** | **1.000** | **1.000** |
| BD-LRU m8 | **1.000** | **1.000** | **1.000** | **1.000** | **1.000** | **1.000** |
| H-LRU m1 (ours) | 0.570 | 0.360 | 0.350 | 0.210 | 0.230 | 0.344 |
| H-LRU m2 | 0.600 | 0.310 | 0.370 | 0.190 | 0.260 | 0.346 |
| H-LRU m3 | 0.610 | 0.320 | 0.400 | 0.210 | 0.320 | 0.372 |
| H-LRU m5 | 0.620 | 0.320 | 0.450 | 0.190 | 0.380 | 0.392 |
| H-LRU m8 | 0.640 | 0.280 | 0.490 | 0.170 | 0.390 | 0.394 |

Table 2: Model performance on permutation composition tasks for different datasets of different sizes: $S_3$ (10k training samples), $S_3$ (250 training samples), $S_4$ (50k training samples), $S_4$ (3k training samples) $S_5$ (100k training samples). The accuracy values reflect the impact of window size (H-LRU) and block size (BD-LRU), both denoted by $m$. We note that BD-LRU performance improves with block size, demonstrating strong sample efficiency by solving the tasks even given limited training data. All models are single-layer configurations with a maximum overall hidden dimension of 6144. See Appendix C for extended table.

challenges for the models, see Table 2. Performance of the H-LRU was found to decrease progressively with increasing group size, indicating a limited capacity for modeling compositional permutations. Increasing the order of recurrence $m$ did not seem to provide any benefits for the performance. We conclude that a strict inductive bias on the structure of the transition matrix prevents H-LRU from solving this task. Moreover, we found that H-LRU is unable to solve our permutation tasks despite having access to negative and complex eigenvalues (see Appendix J for our eigenvalue analysis). This indicates that the presence of such eigenvalues is insufficient for these tasks and highlights that the structure of state mixing plays a more critical role.

In contrast, BD-LRU with moderate block sizes was able to successfully solve all permutation tasks for all group sizes, matching the performance of LSTM and outperforming all other recurrent architectures tested. Importantly, consistent with the previously demonstrated parameter efficiency, BD-LRU with block size 5 also solved the $S_5$ task using as few as 200K parameters, matching the parameter efficiency of more computationally demanding non-linear LSTM model. Furthermore, we found that BD-LRUs are also sample-efficient in learning permutations, outperforming even LSTM in the regime of limited training data. We notice that in our low training token regime Deltaproduct$_4$ fails to learn the $S_5$ dataset. However, when the number of training samples approaches the token counts used in the study Siems et al. (2025), it is capable of solving $S_5$ task, showing that low-rank matrices are less sample-efficient compared to BD-LRU. Our findings align well with our predictions that dense blocks of BD-LRU are well-suited for implementing permutations between hidden states. The consistent improvement with larger block sizes on permutation tasks of increasing complexity highlights the advantage of the inductive bias in BD-LRU architecture.

## 6 IMPLEMENTATION

The parallel scan algorithm in LRNNs allows them to efficiently process long sequences using constant memory and with logarithmic time complexity. Following the classic approach (Blelloch, 1990), we consider a recurrence of the form

$$\mathbf{h}_{i+1} = \begin{cases} \mathbf{b}_0, & \text{if} \quad i = 0 \\ (\mathbf{h}_i \bigotimes_v \mathbf{A}_i) \bigoplus \mathbf{b}_i, & \text{if } 0 \leq i < n \end{cases}, \tag{8}$$

where $\mathbf{h}_i, \mathbf{b}_i \in \mathbb{R}^N, \mathbf{A}_i \in \mathbb{R}^{N \times N}$ and associative operators: $\bigotimes_v$ is matrix-vector multiplication, $\bigotimes_M$ is matrix-matrix multiplication and $\bigoplus$ point-wise vector summation.

Defining following associative operator ● and making substitution to sequence of pairs,

$$HOPscan = \begin{cases} \mathbf{c}_i = [\mathbf{A}_i, \mathbf{b}_i] \\ \mathbf{c}_i \bullet \mathbf{c}_j \equiv [\mathbf{c}_{i,A} \bigotimes_M \mathbf{c}_{j,A}, (\mathbf{c}_{i,b} \bigotimes_v \mathbf{c}_{j,A}) \bigoplus \mathbf{c}_{j,b}] \end{cases}, \tag{9}$$

reduces recurrence 8 to classic prefix sum and allows application of up and down sweeps of the Blelloch scan (For pytorch implementation see Appendix I).

In many modern LRNNs, $\mathbf{A}_i$ is diagonal ($\mathbf{c}_{i,A} \bigotimes_M \mathbf{c}_{j,A} \sim N$), therefore parallel scan 9 enables efficient parallel processing by reducing the time complexity from $NT$ to $N \log(T)$. However, in more general case presented in Eq. 8, parallel scan changes the time complexity from $N^2 T$ to $N^3 \log(T)$. For large dense matrices $\mathbf{A}_i$ amd/or short sequences, this change in complexity is not beneficial due to the high complexity of matrix-matrix multiplication ($\mathbf{c}_{i,A} \bigotimes_M \mathbf{c}_{j,A} \sim N^3$). However, if we exploit the block diagonal structure of the transition matrices in H-LRU and BD-LRU, we can reduce the time complexity of parallel scan from $N^3 \log(T)$ to $Hm^3 \log(T)$, where $m$ is the block size and $H$ is the number of blocks ($Hm = N$). Therefore, for moderate block sizes with $m^2 \ll N$ we can achieve a significant increase in throughput in the parallel scan implementation compared to sequential implementation.

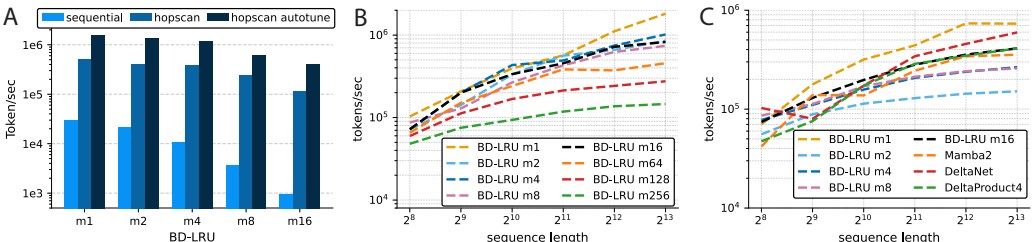

Figure 3: Model throughput on the selective copying task. (**A**) Comparison of sequential, higher-order parallel, and autotuned higher-order parallel implementations of BD-LRUs with 128 blocks and with a sequence length of 2048, illustrating advantage of parallel scan implementation and the trade-off between expressivity and efficiency. BD-LRU is shown for illustration purposes only, but H-LRU employs the same parallel scan implementation and achieves comparable throughput. (**B**) Comparison for layers with hidden size of 768 and accordingly adjusted number of blocks. Note that trade-off between expressivity and efficiency increases over longer sequences. (**C**) Throughput comparison of parameter-matched layers ($\sim$33M parameters). Number of blocks is adjusted to ensure consistent model sizes across architectures. BD-LRU achieves throughput competitive with other LRNN baselines. Notably, larger block sizes demonstrate higher practical efficiency despite increased theoretical complexity, due to superior utilization of GPU hardware operations.

**Parallel scan implementation enables competitive throughput.** In experiments with single-layer models containing 128 blocks and trained on sequences of length 2048, when runtime is less influenced by GPU characteristics and more reflective of algorithmic complexity, we found that increasing block size reduces throughput, revealing the predicted trade-off between expressivity and efficiency, see Fig. 3A. For comparison, we also evaluated models with a fixed hidden size of 768 and adjusted the number of blocks accordingly, see 3B. We found that the expressivity–efficiency trade-off becomes more pronounced as sequence length increases. In particular, block sizes larger than 16 exhibit a substantial decline in throughput at longer sequence lengths.

We also tested models with parameter-matched layers ($\sim$33M parameters), where number of blocks is adjusted to ensure consistent model sizes across architectures, see Fig. 3C. We note that our most efficient implementation relies on compilation with maximal autotuning; thus, the performance differences across block sizes primarily reflect kernel optimization in PyTorch and achieved GPU utilization. We found that certain block sizes align more favorably with GPU architectures, analogous to how specific batch sizes optimize memory utilization. In particular, we found that moderately large block sizes ($m = 16$) demonstrate higher practical efficiency despite increased theoretical complexity, due to superior utilization of GPU hardware operations.

Overall, we observed that our parallel scan implementation offers substantial improvements over sequential implementations, enables BD-LRUs and H-LRUs to achieve throughput comparable to the one of linear baselines, and effectively scales with sequence length.

## 7 REPRODUCIBILITY AND LLM USAGE STATEMENTS

All code used for the simulations performed in this study will be made publicly available (GitHub repo) subject to the acceptance of this work. Code snippets of the critical parts of the implementations are made available in Appendix I. Parts of the text were refined with the assistance of an LLM to improve wording and readability.

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

# A CONCLUSION AND OUTLOOK

We introduced H-LRU and BD-LRU as structured extensions of linear recurrent models that enhance temporal and channel-wise state mixing. Our results show that proper gate normalization is essential for scaling such models with window/block size, that H-LRU excels at parameter-efficient compression, while overall BD-LRU is the best-performing architecture on our benchmark of synthetic tasks, and that our parallel-scan implementation can maintain competitive efficiency of block diagonal architectures. Taken together, our empirical results indicate that the state-mixing structure, rather than width alone, acts as an important driver for improved expressivity in LRNNs.

In our experiments, we observed clear task-dependent differences in how performance scales with block size. Simple tasks such as in-context recall, S3, and Parity are effectively solved with block size 2, nearly eliminating any expressivity–efficiency trade-off. More challenging autoregressive problems such as selective copying, S4, S5, and Regular Languages benefit substantially from larger block sizes. In contrast, the compression auto-encoding task exhibits a distinct scaling pattern: intermediate block sizes achieve the best results, while very large blocks degrade average performance across datasets. We also observe the same scaling behavior in our language modeling experiments, supporting general nature of our findings (see Appendix B).

We also find that H-LRU is particularly effective on compression, likely due to its higher-order recurrence structure, whereas BD-LRU is highly parameter- and sample-efficient on permutation-heavy tasks, consistent with the advantages of dense intra-block mixing. Importantly, both architectures maintain strong throughput on long sequences, making moderate-to-large block sizes viable in practice; however, for very large parameter counts, GPU utilization can become a bottleneck.

Overall, our results indicate that the optimal block or window size $m$ is inherently task-dependent. In practice, we recommend beginning with moderate block/window sizes(with moderate hidden dimension) and adjusting upward or downward based on task complexity, sequence length, and modeling objective, thereby navigating the expressivity–efficiency trade-off. More broadly, the problem of selecting appropriate inductive biases and model scales remains an open research question in machine learning, and we hope that our findings contribute an additional perspective to this ongoing direction of research.

One potential limitation is that our study explored only a subset of the possible parametrizations for the selective gates; a broader investigation could uncover even more effective formulations. Another limitation lies in computational performance; we observed that the throughput of our models degrades more rapidly with increasing batch sizes compared to highly optimized baselines such as Mamba, which presents a clear direction for future engineering efforts. Evaluating the proposed architectures on large-scale language modeling, investigating deeper and hybrid architectures, their generalization to higher-order and block-diagonal SSMs, and, in general, optimizing the implementation to further improve computational efficiency are additional topics left for future studies.

# B  LANGUAGE MODELING

Our language modeling experiments with BD-LRU and H-LRU further corroborate the findings from our synthetic task evaluations, see Fig. 4. By varying the hidden size of BD-LRU, we obtain models in the 140M–210M parameter range, see Fig. 4A. BD-LRU with moderate block sizes achieve the best perplexity, whereas diagonal models ($m = 1$) show early saturation with increasing hidden size. These difference are in good agreement with what we found on the MAD benchmark. Architectures with block sizes between 2 and 4 outperform diagonal networks, while models with 8 and 16 block sizes, despite being theoretically more expressive, underperform in practice. These results indicate that moderate block sizes provide a more effective inductive bias for language modeling, in line with our observations on synthetic tasks

We also conducted language-modeling experiments with H-LRU using configurations matched in parameter count to their BD-LRU counterparts, see Fig. 4B for 140M parameters. Consistent with our synthetic benchmarks, H-LRU exhibits stronger parameter efficiency. However, to match the parameter budget of a BD-LRU, H-LRU requires increasing hidden dimension by a factor of $m$, which in turn reduces throughput and increases memory consumption by approximately the same factor, see 6. For example, H-LRU model with $m = 16$ shown in Figure 4B already occupies 95% of the H100 GPU memory while containing only 140 M parameters. Therefore, although H-LRU is more parameter-efficient, it is substantially more computationally demanding and more costly to scale compared to BD-LRU.

We conduct our experiments on 2.5B tokens from the well-established FineWeb dataset(Penedo et al., 2024) using the PlainLM training setup (Ajroldi, 2024). All models are trained on a single NVIDIA H100 GPU, with the largest configuration utilizing approximately 95% of the device's memory. All models are trained with the AdamW optimizer (Loshchilov and Hutter, 2017) with parameters $\beta_1 = 0.9, \beta_2 = 0.95, \epsilon = 10^{-8}$ and a cosine scheduler (Loshchilov and Hutter, 2016) (max LR 0.003, min LR: $10^{-5}$). Consistent with our throughput analysis 6, we observe that models with larger block sizes achieve higher training throughput for the same parameter count due to better GPU utilization. Overall, our language-modeling results align well with the results observed on synthetic tasks for both architectures.

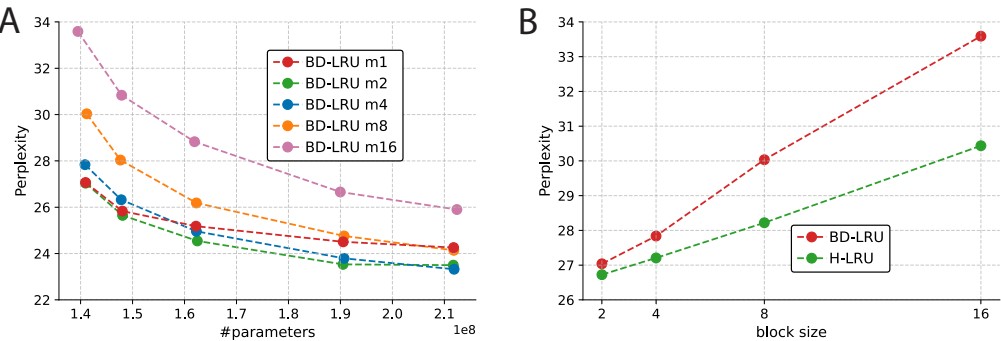

Figure 4: Scaling analysis with hidden size with respect to final perplexity on 2.5B token of FineWeb. All models are trained on a single NVIDIA H100 GPU. **A.** By varying the hidden size of BD-LRU, we obtain models in the 140M–210M parameter range. ***Note that moderate block sizes provide a more effective inductive bias for language modeling.*** **B.** We compare H-LRU and BD-LRU models with 140M parameters. ***Note that matching the parameter budget of a BD-LRU requires increasing the H-LRU hidden dimension by a factor of*** $m$***, making H-LRU substantially more costly to scale.*** For example, shown H-LRU model with $m = 16$ already utilizes 95% of the H100 GPU memory, while BD-LRU with $m = 16$ can be scaled up to 210M parameters with the same memory requirements, see **A.**

| Models | Recall | Copy | Compress | Overall |
|---|---|---|---|---|
| LSTM | **1.000** | **1.000** | 0.750 | 0.916 |
| Mamba2 | **1.000** | 0.807 | 0.720 | 0.842 |
| Deltanet[-1,1] | **1.000** | 0.892 | 0.782 | 0.892 |
| Deltaproduct$_4$[-1,1] | **1.000** | **1.000** | 0.717 | 0.906 |
| BD-LRU m1 (ours) | 0.775 | 0.835 | 0.725 | 0.778 |
| BD-LRU m2 | **1.000** | 0.962 | 0.760 | 0.908 |
| BD-LRU m3 | **1.000** | 0.980 | 0.762 | 0.916 |
| BD-LRU m4 | **1.000** | 0.983 | **0.785** | **0.922** |
| BD-LRU m5 | **1.000** | 0.985 | 0.782 | **0.922** |
| BD-LRU m6 | **1.000** | 0.980 | 0.775 | 0.918 |
| BD-LRU m8 | **1.000** | 0.992 | 0.748 | 0.913 |
| BD-LRU m16 | **1.000** | 0.998 | 0.725 | 0.907 |
| H-LRU m1 (ours) | 0.785 | 0.848 | 0.760 | 0.797 |
| H-LRU m2 | 0.998 | 0.855 | 0.770 | 0.874 |
| H-LRU m3 | **1.000** | 0.855 | 0.772 | 0.876 |
| H-LRU m4 | **1.000** | 0.845 | 0.775 | 0.873 |
| H-LRU m5 | **1.000** | 0.838 | 0.775 | 0.871 |
| H-LRU m6 | **1.000** | 0.818 | 0.775 | 0.864 |
| H-LRU m8 | **1.000** | 0.810 | 0.768 | 0.859 |
| H-LRU m16 | **1.000** | 0.680 | 0.705 | 0.795 |

Table 3: Performance on the in-context recall, selective copying and compression tasks. The presented results are the average of best test accuracies across four configurations of the corresponding synthetic dataset with different vocabulary sizes, sequence lengths and number of training examples. Results are shown for different window (H-LRU) abd block sizes (BD-LRU) $m$. Note that overall performance of our models consistently improves with window/block size up to approximately 3–5, after which the gains saturate or exhibit slight degradation. All models are single-layer configurations with a maximum overall hidden dimension of 6144.

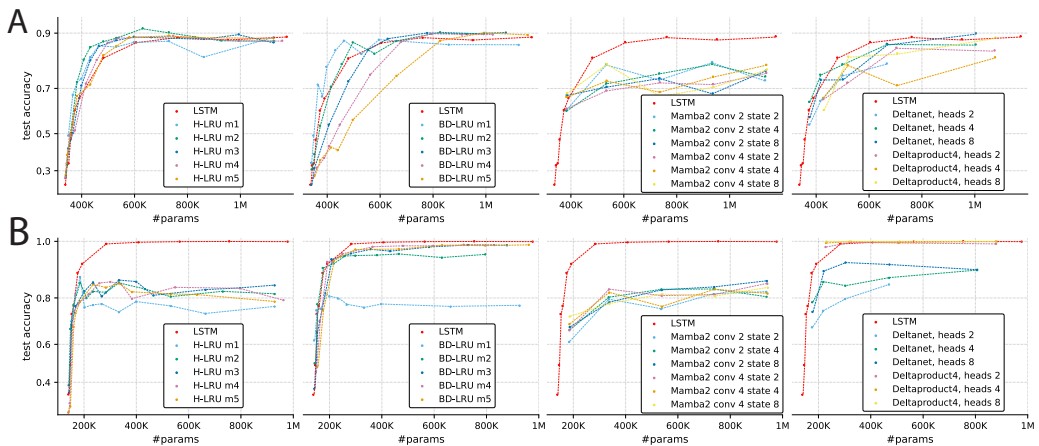

Figure 5: Performance of different single-layer models as a function of the hidden size in the compression task (**A**) and the selective copying task (**B**). Results are shown for different window sizes (H-LRU) and block sizes (BD-LRU) $m$. We compare our networks with different configurations of Mamba (with two sizes of the convolution kernel (2,4) and several values of the state space expansion factor (2,4,8)). For comparison to low-rank models, we also include DeltaNet and DeltaProduct with 4 Householder transforms which have different number of heads (2,4,8).

| Models | $S_3$ (10k samples) | $S_3$ (250) | $S_4$ (50k) | $S_4$ (3k) | $S_5$ (100k) | Overall |
|---|---|---|---|---|---|---|
| LSTM | **1.000** | 0.320 | **1.000** | 0.370 | **1.000** | 0.738 |
| Mamba2 | 0.660 | 0.280 | 0.430 | 0.120 | 0.260 | 0.350 |
| Deltanet[-1,1] | **1.000** | 0.260 | 0.470 | 0.140 | 0.140 | 0.402 |
| Deltaproduct$_4$[-1,1] | **1.000** | 0.270 | **1.000** | 0.130 | 0.140 | 0.508 |
| BD-LRU m1 (ours) | 0.560 | 0.380 | 0.340 | 0.220 | 0.210 | 0.340 |
| BD-LRU m2 | **1.000** | 0.490 | 0.700 | 0.360 | 0.340 | 0.576 |
| BD-LRU m3 | **1.000** | **1.000** | **1.000** | 0.430 | 0.480 | 0.782 |
| BD-LRU m4 | **1.000** | **1.000** | **1.000** | **1.000** | 0.880 | 0.976 |
| BD-LRU m5 | **1.000** | **1.000** | **1.000** | **1.000** | **1.000** | **1.000** |
| BD-LRU m6 | **1.000** | **1.000** | **1.000** | **1.000** | **1.000** | **1.000** |
| BD-LRU m8 | **1.000** | **1.000** | **1.000** | **1.000** | **1.000** | **1.000** |
| BD-LRU m16 | **1.000** | **1.000** | **1.000** | **1.000** | **1.000** | **1.000** |
| H-LRU m1 (ours) | 0.570 | 0.360 | 0.350 | 0.210 | 0.230 | 0.344 |
| H-LRU m2 | 0.600 | 0.310 | 0.370 | 0.190 | 0.260 | 0.346 |
| H-LRU m3 | 0.610 | 0.320 | 0.400 | 0.210 | 0.320 | 0.372 |
| H-LRU m4 | 0.620 | 0.310 | 0.410 | 0.190 | 0.340 | 0.374 |
| H-LRU m5 | 0.620 | 0.320 | 0.450 | 0.190 | 0.380 | 0.392 |
| H-LRU m6 | 0.630 | 0.280 | 0.450 | 0.170 | 0.390 | 0.384 |
| H-LRU m8 | 0.640 | 0.280 | 0.490 | 0.170 | 0.390 | 0.394 |
| H-LRU m16 | 0.660 | 0.260 | 0.510 | 0.160 | 0.390 | 0.396 |

Table 4: Model performance on permutation composition tasks for different datasets of different sizes: $S_3$ (10k training samples), $S_3$ (250 training samples), $S_4$ (50k training samples), $S_4$ (3k training samples) $S_5$ (100k training samples). The accuracy values reflect the impact of window size (H-LRU) and block size (BD-LRU), both denoted by $m$. We note that BD-LRU performance improves with block size, demonstrating strong sample efficiency by solving the tasks even given limited training data. All models are single-layer configurations with a maximum overall hidden dimension of 6144.

# D  EXPERIMENTS

**Synthetic token manipulation tasks.**   We benchmarked our architectures using the Mechanistic Architecture Design (MAD) framework (Poli et al., 2024), a framework for efficient model evaluation and prototyping. The MAD protocol is motivated by the challenge of predicting how architectural choices impact performance at scale. The working hypothesis of MAD is that an architecture's macroscopic scaling behavior can be effectively predicted by its performance on a set of microscopic, mechanistic tasks.

The benchmark consists of a diverse suite of sequence modeling challenges designed to test core token manipulation capabilities. By evaluating models at a small, fixed computational scale, MAD produces a relative ranking of architectures that has been shown to be predictive of their compute-optimal performance in large-scale language modeling (Poli et al., 2024). This approach not only approximates scaling outcomes, but also provides valuable insights into the compositional skills and failure modes of a given design.

In particular, we utilize three tasks from the MAD framework:

- **Compression task**. Models are tasked to compress a random sequence of input tokens into a single aggregation token. Then, this aggregation token is passed through an encoder MLP, the output of which is used to reconstruct the original sequence via a decoder MLP. All models were tested using a standard encoder-decoder architecture (Embedding, Tested Model, MLP Encoder, MLP Decoder).

- **Selective copying task**. Models are tasked with copying tokens from one position of an input sequence to a later position of the sequence, while ignoring irrelevant noise tokens that are randomly inserted into the sequence. This task is designed to evaluate the ability of a model to perform selective temporal integration in the specific order of occurrence in the sequence. All models were tested using a standard decoder-only architecture (Embedding, Tested Model, MLP Decoder).

- **Associative recall task**. Models are presented with an input sequence of key-value pairs and tasked with retrieving all values from the input sequence associated with the presented keys. This task tests the ability of a model to adaptively retrieve information depending on the established in-context associations. All models were tested using a standard decoder-only architecture (Embedding, Tested Model, MLP Decoder).

In our experiments, each model was evaluated across four configurations: a baseline (vocabulary size: 16, sequence length: 64, training examples: 20,000) and three variations designed to probe specific failure modes. These variations all use the same base parameters, but independently (i) increase the vocabulary size to 32, (ii) extend the sequence length to 128, or (iii) reduce the training set to 10,000 examples to test vocabulary handling, long-range capabilities, and sample efficiency, respectively.

**Synthetic permutation tasks.** In our experiments, we employ synthetic datasets derived from the symmetric permutation groups $S_n$, which denotes the group of all possible permutations of $n$ elements. These groups provide a natural hierarchy of complexity: $S_2$ contains only two permutations and is fully commutative, making it relatively simple to model. In contrast, groups with $n \leq 3$ (e.g., $S_3, S_4, S_5$) are non-commutative, and their size grows factorially with $n$, which rapidly increases the difficulty of learning the underlying structure. For instance, $S_3$, with six elements, is the smallest non-commutative group. Geometrically, $S_3$ can be interpreted as the group of symmetries of an equilateral triangle, including both rotations and reflections. The complexity increases substantially with $S_4$, which contains 24 elements and corresponds to the full symmetry group of a regular tetrahedron. $S_4$ introduces more intricate subgroup structures and non-trivial normal subgroups. Extending further, $S_5$ has 120 elements and is the first symmetric group that is not solvable, representing the symmetries of a regular pentagon in the plane.

We assess model performance on the synthetic permutation group task from Merrill et al. (2024), which is designed to probe state-tracking and generalization to complex structures. Using their toolbox, we generated datasets for the symmetric groups $S_3$, $S_4$, and $S_5$ with a fixed sequence length of 16. To evaluate sample efficiency, we created five distinct data configurations: $S_3$ (10k and 250 examples), $S_4$ (50k and 3k examples), and $S_5$ (100k examples). The $S_5$ setting is particularly data-limited compared to the multi-million-example setups used in previous studies (Siems et al., 2025). All models were tested using a standard decoder-only architecture (Embedding, Tested Model, MLP Decoder), consistent with the MAD benchmark protocol.

**Training details.** All models were implemented in PyTorch (Paszke et al., 2019). For training, we follow the experimental settings of the MAD framework. All models are trained with the AdamW optimizer (Loshchilov and Hutter, 2017) with parameters $\beta_1 = 0.9, \beta_2 = 0.999, \epsilon = 10^{-8}$ and a cosine scheduler (Loshchilov and Hutter, 2016) (minimum LR: 0.00001), with the initial learning rate selected from 0.001, 0.0005, 0.0001. The final reported metric is the best test accuracy across all three learning rate configurations and five runs with distinct random seeds. For training we used NVIDIA A100 and NVIDIA H100, while we used NVIDIA H100 for benchmarking the best throughput across models.

## E  COMPUTATIONAL COMPLEXITY

This section provides a breakdown of the Floating Point Operations (FLOPs) required for hidden-to-hidden state transition in the recurrent architectures discussed. For this breakdown, we define the dimension of the hidden state as $H$. The sequence length is denoted as $T$. For Mamba2, the state expansion factor is denoted by $S$. In DeltaNet and DeltaProduct4, $N_h$ denotes the number of heads, $C$ denotes the number of chunks in the DeltaNet implementation, $H_n$ denotes the number of Householder transformations, and $r = 1$ denotes low rank. The calculations focus on the recurrence mechanism, omitting additional components like the input projections or gating, as they can be precomputed in advance. A multiply-add operation is counted as 2 FLOPs.

Table 5: Summary of computational costs for hidden state updates.

| Architecture | FLOPs per recurrent step | Implementation complexity |
|---|---|---|
| LSTM | $8H^2 + 25H$ | $O(TH^2)$ |
| H-LRU | $2Hm + 2H$ | $O(Hm^2 log(T))$ |
| BD-LRU | $2Hm^2 + 2H$ | $O(Hm^2 log(T))$ |
| Mamba2 | $2HS$ | $O(T(H^2 + HS))$ |
| DeltaNet | $N_h(4Hr + 4H)$ | $O(TCH + TH^2)$ |
| DeltaProduct4 | $H_n N_h(4Hr + 4H)$ | $O(H_n(TCH + TH^2))$ |

# F    PROOF OF PROPOSITION 1.

First, note that stability is trivial. We can reason blockwise: assuming $\sum_j |(\mathcal{A}_t^k)_{i,j}| \leq 1$ implies that the eigenvalues of state-transition matrix $\lambda_{i,t}^k \leq 1$. Therefore, the product of such matrices will result in dynamical stability.

Next, by block-diagonality, it is sufficient to show that for all $k \in [1, m]$, $\|\mathbf{h}_T^k\|_\infty \leq \max_{t \in [0,T]} \|\mathbf{v}_t^k\|_\infty$. Let $h_{i,t}^k$ be the $i$-th coordinate of the generic $k$-th block hidden state $\mathbf{h}_t^k$ at time $t$.

$$
\begin{bmatrix} h_{1,t}^k \\ \vdots \\ h_{m,t}^k \end{bmatrix} = \begin{bmatrix} a_{1,1,t}^k & \cdots & a_{1,m-1,t}^k & a_{1,m,t}^k \\ a_{2,1,t}^k & \cdots & a_{2,m-1,t}^k & a_{2,m,t}^k \\ \vdots & \ddots & \vdots & \vdots \\ a_{m,1,t}^k & \cdots & a_{m,m-1,t}^k & a_{m,m,t}^k \end{bmatrix} \times \begin{bmatrix} h_{1,t-1}^k \\ \vdots \\ h_{m,t-1}^k \end{bmatrix} + \begin{bmatrix} a_{1,0,t}^k \\ \vdots \\ a_{m,0,t}^k \end{bmatrix} \odot \begin{bmatrix} v_{1,t}^k \\ \vdots \\ v_{m,t}^k \end{bmatrix}. \tag{10}
$$

Hence,

$$
h_{i,t}^k = \sum_{j=1}^m a_{i,j,t}^k h_{j,t-1}^k + a_{i,0,t}^k v_{i,t}^k. \tag{11}
$$

It is then clear that by subadditivity of the absolute value,

$$
|h_{i,t}^k| \leq \sum_{j=1}^m |a_{i,j,t}^k| \cdot |h_{j,t-1}^k| + |a_{i,0,t}^k| \cdot |v_{i,t}^k|. \tag{12}
$$

Hence, by collecting the non-coefficient terms, we find a further upper bound

$$
|h_{i,t}^k| \leq \left( \sum_{j=1}^m |a_{i,j,t}^k| + |a_{i,0,t}^k| \right) \cdot \max \left[ |v_{i,t-1}^k|, \max_{j \in [1,m]} |h_{j,t}^k| \right]. \tag{13}
$$

By hypothesis, $\sum_{j=1}^m |a_{i,j,t}^k| + |a_{i,0,t}^k| = \sum_j |(\mathcal{A}_t^k)_{i,j}| \leq 1$, and hence we conclude that

$$
|h_{i,t}^k| \leq \max \left[ |v_{i,t}^k|, \max_{j \in [1,m]} |h_{j,t-1}^k| \right]. \tag{14}
$$

At this point, we can finalize the proof by induction. We want to show that $\|\mathbf{h}_T^k\|_\infty \leq \max_{t \in [0,T]} \|\mathbf{v}_t^k\|_\infty$. Let us start from $T = 1$. Since $h_{i,0}^k = 0$ for all $i \in [1, m]$, we have

$$
h_{i,1}^k = a_{i,0,t}^k v_{i,1}^k, \tag{15}
$$

hence, again because $\sum_j |(\mathcal{A}_0^k)_{i,j}| \leq 1$, $|h_{i,1}^k| \leq |v_{i,1}^k|$, we can conclude that $\|\mathbf{h}_1^k\|_\infty \leq \|\mathbf{v}_1^k\|_\infty$. Let us then assume by induction that $\|\mathbf{h}_{T-1}^k\|_\infty \leq \max_{t \in [0,T-1]} \|\mathbf{v}_t^k\|_\infty$. Recall that by Equation 14,

$$|h_{i,t}^k| \leq \max \left[ |v_{i,t}^k|, \max_{j \in [1,m]} |h_{j,t-1}^k| \right] \tag{16}$$

$$= \max \left[ |v_{i,t}^k|, \|\mathbf{h}_{t-1}^k\|_\infty \right]. \tag{17}$$

Hence,

$$\|\mathbf{h}_t^k\|_\infty = \max_{j \in [1,m]} |h_{j,t}^k| \tag{18}$$

$$\leq \max_{j \in [1,m]} \max \left[ |v_{i,t}^k|, \|\mathbf{h}_{t-1}^k\|_\infty \right] \tag{19}$$

$$= \max \left[ \max_{j \in [1,m]} |v_{i,t}^k|, \|\mathbf{h}_{t-1}^k\|_\infty \right] \tag{20}$$

$$\leq \max \left[ \|\mathbf{v}_t^k\|_\infty, \max_{t \in [0,T-1]} \|\mathbf{v}_t^k\|_\infty\| \right] \tag{21}$$

$$= \max_{t \in [0,T]} \|\mathbf{v}_t^k\|_\infty, \tag{22}$$

where in the second-last line we used the induction hypothesis.

## G  SELECTIVITY ABLATION

To isolate and quantify the contribution of selectivity, we conducted an ablation study. In this analysis, the input-dependent selective gates in both the H-LRU and BD-LRU architectures were replaced with data-invariant, learnable parameters.

As hypothesized, the non-selective variants exhibited a significant performance degradation compared to their selective counterparts on our synthetic benchmark. On tasks requiring dynamic token manipulation—such as in-context recall, selective copying, and permutation composition—the non-selective models failed to achieve meaningful performance. For these tasks, increasing the window or block size yielded no discernible improvement, confirming the necessity of selectivity.

However, the results on the compression task were more nuanced, see Fig. 6. We observed that our proposed $L1$ normalization scheme enabled the non-selective models to improve with larger block and window sizes, albeit at a lower rate than their selective analogs.

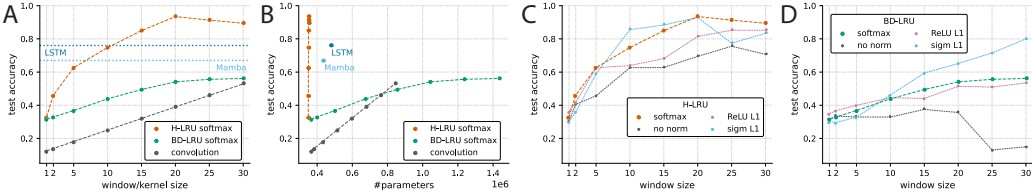

Figure 6: Scaling analysis of non-selective models on the compression task. **A**. Performance as a function of window size $m$ of non-selective higher-order LRU (H-LRU) and block size $m$ of block diagonal LRU (BD-LRU). For the convolutional baseline, the performance presented as a function of kernel size. **B**. The same results plotted against parameter count. Note that scaling with window size of non-selective H-LRU demonstrates extreme parameter efficiency, resulting in a nearly vertical trajectory on the plot. **C**. Comparison of scaling properties between different parameterizations for H-LRU. **D**. Comparison of scaling properties between different parameterizations for BD-LRU.

To highlight the advantages of recurrent architectures, we used a convolution layer as a baseline. This model is limited to explicit, local time mixing within its kernel, in contrast to the implicit and unbounded temporal integration provided by a hidden state. Our experiments showed that H-LRU decisively outperforms the convolution on the compression task. This demonstrates the critical role

of recurrent state mixing for tasks requiring efficient long-range temporal reasoning. Furthermore, the non-selective H-LRU with large window sizes ($m > 15$) demonstrated strong performance, surpassing the LSTM and Mamba baselines and even approaching the performance of our selective models. This finding underscores the powerful inductive bias of the higher-order recurrence for parameter-efficient compression.

In contrast, the non-selective BD-LRU performed poorly on the compression task, only marginally surpassing the convolution baseline. Interestingly, for this non-selective variant, the sigmoidal $L1$ normalization outperformed softmax normalization, highlighting a difference in how these schemes interact with selective versus fixed parameterizations.

In addition, when we analyzed H-LRU with minimal point-wise selective gates which don't mix channel dimensions, we observed very moderate improvement in compression task. This indicates that not only selectivity itself but also density of selectivity in gates plays important role in improving networks' expressivity.

While the overall performance of these non-selective models is modest, their parameter efficiency can become advantageous in resource-constrained settings. Given the strong compression results of the non-selective H-LRU, we hypothesize that such models could be optimized for use as highly efficient embedding layers, a direction we leave for future research.

## H  RELATION BETWEEN EXPRESSIVITY OF LRUs AND STATE SPACE DUALITY

Recently, it has been shown that there is a direct correspondence between state space models, the Transformer architecture and structured attention matrices Dao and Gu (2024). Following this approach, we can reformulate the general LRU as a general discrete time SSM

$$
\begin{aligned}
\mathbf{h}_t &= \mathbf{A}_t \times \mathbf{h}_{t-1} + \mathbf{B}_t \times \mathbf{v}_t \\
\mathbf{y}_t &= \mathbf{C}_t \times \mathbf{h}_t.
\end{aligned}
\tag{23}
$$

Here, we consider the general case of SSMs, in which mixing matrices $\mathbf{C}_t, \mathbf{A}_t, \mathbf{B}_t$ are dense matrices. We note that although state space models are commonly defined in continuous time, they have to be discretized for implementation, at which point they conform to the discrete form described by Eq. 23. In this study, we effectively ignored the role of $\mathbf{C}_t$, but it can be introduced without affecting the validity of our arguments.

Following the approach of reformulating state space models (SSMs) as attention mechanisms, the architecture given in Eq. 23 can be expressed in block matrix representation assuming a fixed sequence length $T$:

$$
\begin{bmatrix} \mathbf{y}_1 \\ \mathbf{y}_2 \\ \mathbf{y}_3 \\ \vdots \\ \mathbf{y}_T \end{bmatrix} = \begin{bmatrix} \mathbf{C}_1\mathbf{B}_1 & \mathbf{0} & \mathbf{0} & \cdots & \mathbf{0} \\ \mathbf{C}_2\mathbf{A}_1\mathbf{B}_1 & \mathbf{C}_2\mathbf{B}_2 & \mathbf{0} & \cdots & \mathbf{0} \\ \mathbf{C}_3\mathbf{A}_2\mathbf{A}_1\mathbf{B}_1 & \mathbf{C}_3\mathbf{A}_2\mathbf{B}_2 & \mathbf{C}_3\mathbf{B}_3 & \cdots & \mathbf{0} \\ \vdots & \vdots & \vdots & \ddots & \vdots \\ \mathbf{C}_T\prod_{j=1}^{T}\mathbf{A}_j\mathbf{B}_1 & \mathbf{C}_T\prod_{j=2}^{T}\mathbf{A}_j\mathbf{B}_2 & \cdots & \cdots & \mathbf{C}_T\mathbf{B}_T \end{bmatrix} \begin{bmatrix} \mathbf{v}_1 \\ \mathbf{v}_2 \\ \mathbf{v}_3 \\ \vdots \\ \mathbf{v}_T \end{bmatrix}
$$

If we abstract the details of SSMs matrices, we obtain the generalized attention formulation:

$$
\begin{bmatrix} \mathbf{y}_1 \\ \mathbf{y}_2 \\ \mathbf{y}_3 \\ \vdots \end{bmatrix} = \begin{bmatrix} \overline{\mathbf{A}}_{1,1} & \mathbf{0} & \mathbf{0} & \cdots & \mathbf{0} \\ \overline{\mathbf{A}}_{2,1} & \overline{\mathbf{A}}_{2,2} & \mathbf{0} & \cdots & \mathbf{0} \\ \overline{\mathbf{A}}_{3,1} & \overline{\mathbf{A}}_{3,2} & \overline{\mathbf{A}}_{3,3} & \ddots & \vdots \\ \vdots & \vdots & \vdots & \ddots & \vdots \end{bmatrix} \begin{bmatrix} \mathbf{v}_1 \\ \mathbf{v}_2 \\ \mathbf{v}_3 \\ \vdots \end{bmatrix}.
\tag{24}
$$

Importantly, elements $\overline{\mathbf{A}}_{k,l}$ of the block attention matrix are matrices as well in this representation. According to State Space Duality Dao and Gu (2024), both the attention in Transformers and diagonal SMMs result in diagonal matrices $\overline{\mathbf{A}}_{k,l}$. So, their architecture allows for efficient parallelization as it separates temporal mixing from channel mixing.

In contrast to diagonal SSMs and LRUs, both H-LRU and BD-LRU architectures result in block-diagonal matrices $\overline{\mathbf{A}}_{k,l}$, allowing richer but limited by block channel mixing inside the generalized

block attention matrix 24. Such channel mixing allows for the state mixing patterns that are not accessible to one layer of diagonal LRU or SSMs. Although the channel mixing in H-LRU is more expressive than the one in a diagonal LRU, it is still more restricted compared to BD-LRU (it is equivalent to mixing only in one row of block-diagonal matrix), placing expressivity of H-LRU between diagonal LRU and BD-LRU. Notably, if we extend SSMs with higher-order or block-diagonal structures, their expressivity would lag behind analogous LRUs due to the restrictions on mixing patterns imposed by the chosen discretization scheme. Overall, the generalized block attention formulation 24 reveals that for both LRUs and SSMs, diagonal, higher-order, block diagonal and dense variants form a hierarchy of architectures, each providing access to increasingly complex state mixing patterns which result in increased expressivity.

# I  CODE SNIPPETS

Following the approach for diagonal LRNNs (Sarnthein, 2025), we implement forward and backward pass for block-diagonal recurrence based on associative scan in PyTorch.

```python
import torch
from torch.autograd.function import Function, FunctionCtx
from torch._higher_order_ops.associative_scan import associative_scan

# helper function to implement reverse mode
def shift(input, shifts, fillval=0):
    # torch.roll without the copy of the wrap-around section
    if shifts > 0:
        output = torch.cat([torch.full_like(input[:, :shifts,...], fillval),
                            input[:, :-shifts,...]], dim=1)
    if shifts < 0:
        output = torch.cat([input[:, -shifts:,...],
                            torch.full_like(input[:, shifts:,...], fillval)], dim=1)
    return output

# Forward pass of associative scan
def scan_hop_fwd(inputs:torch.Tensor, coeffs:torch.Tensor, reverse=False):

    # Higher-Order Op Implementation
    def op(acc:dict, curr:dict):
        c = torch.einsum('bcij,bcjk->bcik',curr['c'],acc['c'])
        x = curr['x'] + torch.einsum('bcij,bcj->bci',curr['c'],acc['x'])
        return dict(x=x, c=c)

    outputs = associative_scan(op, dict(x=inputs, c=coeffs), dim=1,
                                 reverse=reverse, combine_mode='generic')['x']
    return outputs

# Backward pass that uses forward pass in reverse mode
def scan_hop_bwd(d_outputs:torch.Tensor, coeffs:torch.Tensor,
                 outputs:torch.Tensor, reverse=False):
    coeffs_bwd = shift(coeffs, -1 if not reverse else 1, fillval=0).permute(0,1,2,4,3)
    d_inputs = scan_hop_fwd(inputs=d_outputs, coeffs=coeffs_bwd, reverse=(not reverse))
    d_coeffs = torch.einsum('btci,btck->btcik',d_inputs,
                            shift(outputs, shifts=1 if not reverse else -1, fillval=0))
    return d_inputs, d_coeffs

# Autograd wrapper
class ScanHopFn(Function):
    @staticmethod
    def forward(ctx:FunctionCtx, inputs:torch.Tensor,
                coeffs:torch.Tensor, reverse:bool=False) -> torch.Tensor:
        outputs = scan_hop_fwd(inputs=inputs, coeffs=coeffs, reverse=reverse)
        ctx.save_for_backward(coeffs, outputs)
        ctx.reverse = reverse
        return outputs

    @staticmethod
    def backward(ctx:FunctionCtx, d_outputs:torch.Tensor):
        coeffs, outputs = ctx.saved_tensors
        d_inputs, d_coeffs = scan_hop_bwd(d_outputs=d_outputs, coeffs=coeffs,
                                          outputs=outputs, reverse=ctx.reverse)
        return d_inputs, d_coeffs, None

# Scan function
def hopscan(inputs:torch.Tensor, coeffs:torch.Tensor):
    return ScanHopFn.apply(inputs, coeffs)
```

Simplified version of H-LRU with autotuned higher-order parallel scan

```python
import torch
import torch.nn.functional as F
import torch.nn as nn
from scans.hopscan import hopscan

@torch.compile(mode="max-autotune", dynamic=False)
class HLRU(nn.Module):
    def __init__(
        self,
        input_dim: int,
        window_dim: int = 4,
        hidden_dim: int = 64,
        **kwargs
    ):
        super().__init__()
        self.input_dim = input_dim
        self.hidden_dim = hidden_dim
        self.window_dim = window_dim
        # initialize projections and gates
        self.proj_gates = nn.Linear(self.input_dim, self.hidden_dim*(self.window_dim+1),
                                    bias=True)
        self.proj_v = nn.Linear(self.input_dim, self.hidden_dim,
                                bias=False)
        self.proj_out = torch.nn.Linear(self.hidden_dim*self.window_dim, self.input_dim,
                                        bias=False)
        # structred 1-off diagonal matrix for companion form
        self.register_buffer("A_temp", torch.diag(torch.ones(self.window_dim-1), 1))

    def forward(self,
        x: torch.Tensor,
        *args, **kwargs
    ):
        """
        x (torch.Tensor): tensor of shape (B T N)
        y (torch.Tensor): tensor of shape (B T N)
        """
        B, T, _ = x.size()
        # projection of input to hidden size
        v = self.proj_v(x) # B T H
        # projections that form selective state gates and input gates
        gates = self.proj_gates(x) # B T H*(m+1)
        gates = gates.reshape(B,T,self.hidden_dim,self.window_dim+1)

        # softmax normalization of coeff A and a_0
        A_t = torch.softmax(gates,-1) # B T H m+1
        # apply gate to input a_0*v
        a0v = A_t[:,:,:,-1:]*v[:,:,:].unsqueeze(-1) # B T H
        # gated input is padded with zeros to get structured form
        a0v = F.pad(a0v,(0,self.window_dim-1)) # B T H m
        # pad A_t to get block diagonal form
        A_t = F.pad(A_t[:,:,:,:-1].unsqueeze(-1),(0,self.window_dim-1))
        # in order to get companion form
        # we add A_temp which is structred 1-off diagonal matrix
        A_t = self.A_temp + A_t # B T H m m

        # parallel scan
        # takes (B T H m) and (B T H m m) and returns (B T H m)
        y=hopscan(a0v, A_t) # B T H m

        # reshape and project back
        y=y.reshape(B,T,self.hidden_dim*self.window_dim) # B T H*m
        y=self.proj_out(y) # B T N
```

Simplified version of BD-LRU with autotuned higher-order parallel scan

```python
import torch
import torch.nn.functional as F
import torch.nn as nn
from scans.hopscan import hopscan

@torch.compile(mode="max-autotune", dynamic=False)
class BDLRU(nn.Module):
    def __init__(
        self,
        input_dim: int,
        window_dim: int = 4,
        hidden_dim: int = 64,
        **kwargs
    ):
        super().__init__()
        self.input_dim = input_dim
        self.hidden_dim = hidden_dim
        self.window_dim = window_dim
        # initialize projections and gates
        self.proj_gates = nn.Linear(self.input_dim,
                                    self.hidden_dim*self.window_dim*(self.window_dim+1),
                                    bias=True)
        self.proj_v = nn.Linear(self.input_dim, self.hidden_dim*self.window_dim,
                                bias=False)
        self.proj_out = torch.nn.Linear(self.hidden_dim*self.window_dim, self.input_dim,
                                        bias=False)

    def forward(self,
        x: torch.Tensor,
        *args, **kwargs
    ):
        """
        x (torch.Tensor): tensor of shape (B T N)
        y (torch.Tensor): tensor of shape (B T N)
        """
        B, T, _ = x.size()
        # projection of input to hidden size
        v = self.proj_v(x) # B T H*m
        # projections that form selective state gates and input gates
        gates = self.proj_gates(x) # B T H*m*(m+1)
        gates = gates.reshape(B,T,self.hidden_dim,self.window_dim,self.window_dim+1)

        # softmax normalization of coeff A and a_0
        A_t = torch.softmax(gates,-1) # B T H m m+1
        # apply gate to input a_0*v
        a0v = A_t[:,:,:,:,-1]*v[:,:,:,:]  # B T H m
        # state-transition matrix
        A_t = A_t[:,:,:,:,:-1] # B T H m m

        # parallel scan
        # takes (B T H m) and (B T H m m) and returns (B T H m)
        y=hopscan(a0v, A_t) # B T H m

        # reshape and project back
        y=y.reshape(B,T,self.hidden_dim*self.window_dim) # B T H*m
        y=self.proj_out(y) # B T N
```

# J   EIGENVALUE ANALYSIS

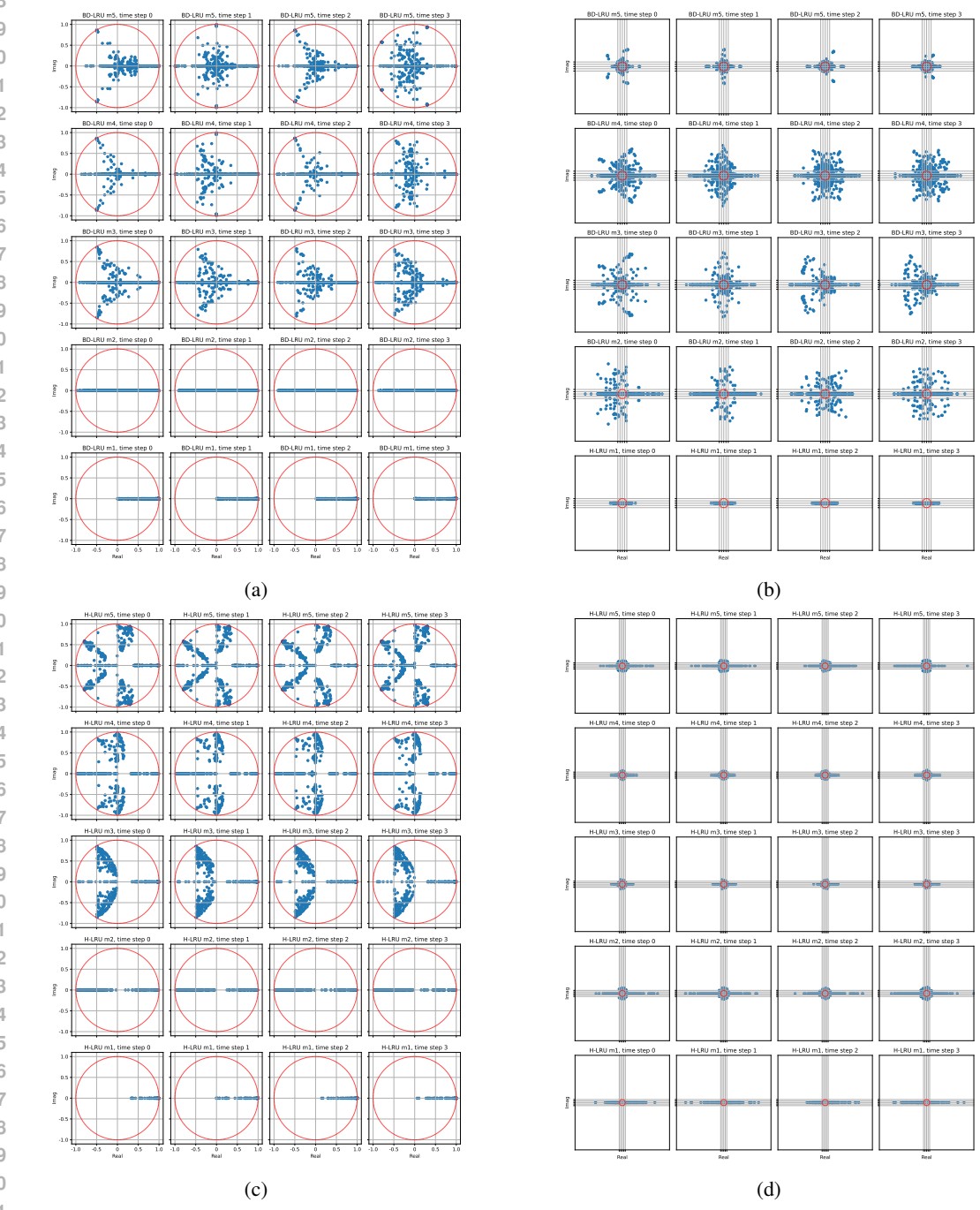

Figure 7: Eigenvalues of LRUs on $S5$ dataset. (a) BD-LRU with softmax normalization. (b) BD-LRU without normalization. (c) H-LRU with softmax normalization. (d) H-LRU without normalization. Each subplot corresponds to a specific time step (horizontal axis) and block size (vertical axis). Models without normalization exhibit unstable transition matrices. Note that as block size increases, the number of available symmetries increases as well.

# K  CHOMSKY HIERARCHY TASKS

The Chomsky hierarchy formalizes increasing levels of expressiveness and computational complexity of formal languages into several hierarchical classes (Chomsky, 1956; Delétang et al., 2022). Here, we tested several tasks from this hierarchy: Parity, Cycle Navigation, Modular Arithmetic with and without brackets. Parity task requires computing whether given binary string is even or not. Cycle Navigation requires computing the end position given a sequence of movements on a cycle of length 5. Modular Arithmetic tasks require computing the result modulo 5 for given sequence of numbers in $(0, 1, 2, 3, 4)$ and operations in $(+, -, \cdot)$, with or without brackets.

In our experiments, we observe that similar to $S_3$ task, Parity task can be solved by BD-LRU with access to negative eigenvalues ($m \geq 2$). For Cycle Navigation task we obtain similar results as for $S_5$ task. BD-LRU is able to solve it starting from $m = 5$. Therefore, the results on these two tasks from Chomsky Hierarchy support our previously found advantage of BD-LRUs on permutations tasks.

Modular arithmetic tasks present a challenge for highly parallel Transformer architecture, often require grokking and having pure generalization (Gromov, 2023). In contrast, it has been shown that sequential nature of state mixing in RNNs has a strongly beneficial bias for arithmetic-like induction (Merrill and Sabharwal, 2023). However, both our linear variants and other modern LRNNs struggle with such arithmetic tasks (Siems et al., 2025), supporting the idea that nonlinearity of state transitions is crucial in such tasks (Chang and Bisk, 2024). In our experiments, we found that BD-LRU were able to solve Modular Arithmetic without brackets, while the version with brackets remained challenging, similar to other RNNs.

| Models | cycle nav | mod arith no brack | mod arith w brack | parity |
|---|---|---|---|---|
| LSTM | **1.000** | 0.976 | 0.663 | **1.000** |
| BD-LRU m1 | 0.434 | 0.370 | 0.370 | 0.512 |
| BD-LRU m2 | 0.425 | 0.493 | 0.417 | **1.000** |
| BD-LRU m3 | 0.597 | 0.546 | 0.434 | **1.000** |
| BD-LRU m4 | 0.608 | 0.459 | 0.435 | **1.000** |
| BD-LRU m5 | **1.000** | 0.525 | 0.422 | **1.000** |
| BD-LRU m6 | **1.000** | 0.433 | 0.440 | **1.000** |
| BD-LRU m8 | **1.000** | 0.553 | 0.395 | **1.000** |
| BD-LRU m16 | **1.000** | **1.000** | 0.448 | **1.000** |

Table 6

