# OpenReview forum: "Improved state mixing in higher-order and block diagonal linear recurrent networks"
_ICLR.cc/2026/Conference — Submitted to ICLR 2026_

### Official Review · Reviewer_E3FU · 2025-10-30

**Soundness:** 2
**Presentation:** 2
**Contribution:** 1
**Rating:** 2
**Confidence:** 4

**Summary:**

Background: traditional recurrent NNs (e.g. LSTMs) rely on fully dense transition matrices.  Linear recurrent NNs (and state space models) are a class of neural networks in which the transition matrix is a diagonal matrix, and they are much faster to compute over long sequences.

The authors introduce and evaluate two new classes of recurrent neural networks (RNNs).  Higher-order linear recurrence (H-LRU) can refer not just to the hidden state at the previous time step, but to prior time steps as well.  Block-diagonal recurrence (BD-LRU) uses a state transition matrix that is block-diagonal, rather than diagonal, and thus offers a compromise between linear RNNs and fully dense RNNs.

The authors show that H-LRUs are actually a type of BD-LRU with a particular block structure.  They then show that BD-LRUs can be tamed by use of a normalization scheme, which normalizes the transition matrix to avoid exploding or vanishing activations and gradients.

The authors compare their RNNs with various block sizes against LSTMs, Mamba, and Deltanet, on several synthetic tasks.

**Strengths:**

For the most part the paper is clear and well-written, and the mathematics are well-presented.

The normalization technique in equation (6) seems valuable.

**Weaknesses:**

The concepts are not hard, but the writing style is rather dense and difficult to slog through.

The major weakness of the paper is (1) extremely small scale, and (2) reliance solely on toy synthetic problems.

In general, I think that synthetic tasks are actually quite useful -- they allow you to evaluate precisely which tasks a given NN architecture is good at.  However, any proposed architecture should *also* be evaluated on real-world data sets, such as one of the many language-modeling datasets.

This problem is exacerbated by the extremely small scale of the NNs involved.  The authors test block sizes for the BD-LRU of between 1 and 5, which is miniscule.  Modern TPU and GPU hardware is typically efficient only for matmuls of dimension 128 or greater.

The final issue is that the actual performance of BD-LRU is bizarre.  In Figure 1, it is simultaneously the best performing and the worst-performing model.  Moreover the performance does not scale with block size as one would expect -- the points seem like a random scatter plot, which looks to me like an instability of the technique.  At the very least, this issue needs to be explained, and the authors do not do so.

The parallel scan algorithm also needs to be properly explained.  A citation to an old paper on prefix sums is not sufficient for equation (8), which is clearly not a simple sum.

**Questions:**

What is your explanation for the results in Figure 1?

In figure 2, where is relu?  Why does relu fail so abysmally?

---

> ### Author Response · Authors · 2025-11-26
>
> We thank the reviewer for their valuable feedback. We are glad that overall they found our text to be well-written and that they agree that the proposed normalization is novel. In the following, we address the main questions raised by the reviewer and hope that these clarifications will allow the reviewer to reassess their evaluation.
>
> **1) Language modeling.** We agree with the reviewer that synthetic tasks are highly informative for evaluating specific capabilities. Moreover, we believe that our study demonstrates how such tasks can effectively reveal advantages and disadvantages of different architectures already at smaller scales. However, we acknowledge that including a standard language modeling benchmark would strengthen our contribution. In response, we have added a new section (Appendix B) presenting language modeling experiments on the FineWeb dataset. Overall, these results further support the main findings of our work (see Appendix~A,B).
>
> **1) Hidden size.**
> We would like to clarify a possible misunderstanding: the block size $m$ is not the hidden size of our block-diagonal recurrent networks. The hidden size is given by the total state dimension, the number of blocks multiplied by the block size $m$. A small block size therefore does not imply a small model.
>
> For context, modern diagonal recurrent architectures correspond to the extreme case of effective block size $m=1$. However, such a small block size is precisely what enables these architectures to scale efficiently to hidden dimensions in the thousands in large-scale models. In our original submission, we used BD-LRU models that reach hidden sizes of up to 1536 and H-LRU sizes of up to 2560. For comparison, GPT-2 layers use hidden sizes between 768 and 1600, and a recent NVIDIA study [1] comparing Mamba and Transformer models evaluates 8B-parameter models with hidden size 4096. To nevertheless address this concern, we have extended our results to hidden sizes of up to 6k for both architectures.
>
> We have also updated our throughput analysis to cover a broader range of block sizes and to illustrate which configurations are most efficient on modern GPUs. To ensure a fair comparison, we present two plots, one showing models with the same hidden size across models and and one with the same parameter budget across models.
>
> **3) Parallel scan implementation.** We thank the reviewer for their feedback on the Implementation section. We have revised the description of our parallel scan algorithm to enhance clarity and now describe all important details that were missing before.
>
> In the new version of the Implementation section, we present an improved description of the recurrence relation that is computed by our algorithm. Importantly, we also add the description of the associative operator and the substitution that we use to reduce our recurrence relation to a prefix sum, which makes application of Blelloch scan possible.
>
> **4) Questions**
> 1. Figure 1 presents the best performing models across all hidden sizes tested. The seeming discrepancy in performance is explained by the fact that the dots represent aggregate scores for models with different hidden dimensions, resulting in cases in which a model with a smaller window can have larger overall FLOPs. We agree with the reviewer that our Figure 1 was hard to understand, and we have updated it in the hope that it makes things clearer. To specifically address the question raised here, we have also added plots that compare different block sizes separately.
>
> 2. Models with ReLU normalization underperformed for several reasons. First, exponential parametrizations have advantages over linear parameterizations in gradient based learning, as was shown previously [2]. This was mentioned in the original submission at the end of paragraph "Normalization allows scaling with window size". Second, in our experiments, we found that models using the ReLu normalization tend to produce transition matrices with zero eigenvalues. To nevertheless address this question, we improved the stability of the ReLU normalization by increasing the epsilon term in the denominator of the L1 normalization, mitigating to a certain extent this degeneracy problem. As anticipated, even with this adjustment, ReLU normalization continues to underperform relative to softmax and sigmoid normalizations. We have updated Figure 2 to include these new results. We hope that this addresses the reviewer's comment.
>
> References:
>
> [1] Waleffe, R., Byeon, W., Riach, D., Norick, B., Korthikanti, V., Dao, T., ... \& Catanzaro, B. (2024). An empirical study of mamba-based language models. arXiv preprint arXiv:2406.07887.
>
> [2] Orvieto, A., Smith, S. L., Gu, A., Fernando, A., Gulcehre, C., Pascanu, R., \& De, S. (2023). Resurrecting Recurrent Neural Networks for Long Sequences. arXiv preprint arXiv:2303.06349.

---

> > ### Author Response · Authors · 2025-12-02
> >
> > We hope that our prior response and the updated revision have thoroughly addressed the concerns raised. We remain open to any further discussion should additional points require clarification.

---

### Official Review · Reviewer_nE3j · 2025-10-31

**Soundness:** 4
**Presentation:** 4
**Contribution:** 2
**Rating:** 2
**Confidence:** 3

**Summary:**

The author introduces that LRNNs face a fundamental tradeoff: diagonal architectures are computationally efficient but expressively limited, while dense architectures are more expressive but computationally expensive. This paper explores structured approaches to bridge this efficiency-expressivity gap.

The author proposed Higher-order Linear Recurrent Units, which mix multiple past hidden states and Block-Diagonal LRUs, where the recurrent weight matrix is block instead of pure full or diagonal. The author notice and proposed a method to solve the stability issue. The author provides experiments to demonstrate the effectiveness of the proposed architecture.

**Strengths:**

The presentation of the architecture is clear, and easy to follow. The notations and formulas are well defined which makes the reading smooth.
The proposed method is novel which has not been proposed before.
There are extensive experiments to evaluate the proposed architecture.

**Weaknesses:**

See Questions

**Questions:**

The author introduces that LRNNs face a fundamental tradeoff: diagonal architectures are computationally efficient but expressively limited, while dense architectures are more expressive but computationally expensive. This paper explores structured approaches to bridge this efficiency-expressivity gap.

The author proposed Higher-order Linear Recurrent Units, which mix multiple past hidden states and Block-Diagonal LRUs, where the recurrent weight matrix is block instead of pure full or diagonal. The author notice and proposed a method to solve the stability issue. The author provides experiments to demonstrate the effectiveness of the proposed architecture.



The presentation of the architecture is clear, and easy to follow. The notations and formulas are well defined which makes the reading smooth.
The proposed method is novel which has not been proposed before.


1. Line 13-14: "Dense and/or nonlinear architectures (e.g., LSTMs) on the other hand are provably more expressive"
    - For architecture with nonlieanrity, no matter the recurrent matrix is full or diagonal, the architecture should have universal approximation property. Why the author state that dense architectures are more expressiv, what is the "expressive" here refers to?

2. Line 876 - 917 Appendix F discuss the expressivity.
    - Firstly, without non linearity it is obvious that diagonal matrices is contained in the block diagonal matrices is contained in the full matrices.
    - Consider a target is in the form $y=C A^n B$, say if A is diagonalizable, then whether the model uses a diagonal or dense matrices does not matter. And in reality a random matrix $A$ has probability 1 to be diagonalizable in complex space. So there expensiveness are very "near" except for a measure zero set of targets.

3. Are there any new trade-off arises in H-LRU? The $m$ determines the window size the hidden state looks back. When increasing this $m$, the memory can be reduced. but as a side effect, the temporal information is also lost. Consider the extreme case where $m$ equals to the sequence length. The the recurrent model would collapse to something like a feedforward network, where the input dimension is the sequence length. This makes is able to look at the entire sequence at once but also lost all the temporal information. Thus, the question is that is there a fixed good $m$ that balances this tradeoff.

4. For BD-LRU, the $m$ determines the number of block and it is fixed. The question of how $m$ should be determined is also a question. If there is no algorithm that can determin this $m$ without actually train the model, I think the significance of the results is much lower.


6. If there are no reparameterzation involved in the implementation as shown in Appendix G, I doubt whether the model can have better performance than S4 or S4D models.  The paper says that the BD-LRU and H-LRU is more expressive than diagonal models, but experiments does not has diagonal models involved.

7. From the experiment the H-LRU does not seems have better results than other architectures. some cases worse than LSTM.

---

> ### Author Response · Authors · 2025-11-26
>
> We thank the reviewer for their valuable feedback. We are pleased that they found the presentation clear, the experimental evaluation thorough, and the proposed approach novel. While we were somewhat surprised by the final score given the largely positive assessment, we appreciate the reviewer's engagement with our work. In the following, we address the main questions raised and hope that these clarifications will allow the reviewer to reassess their evaluation.
>
> **1. \& 2. Questions on expressivity.** In this paper, we study block-diagonal linear time-variant RNNs with input-dependent transition matrices. It is well known [1,2] that diagonal input-dependent linear RNNs have reduced expressivity~(we develop on what we mean below) compared to dense input-dependent linear RNNs. The reviewer correctly states that dense real matrices are diagonalizeable with probability 1. This is correct, however (as discussed thoroughly in [2]), this is not enough in the time-variant case: All $A_t$s need to be simultaneously diagonalizable for the system to enjoy the diagonal-dense equivalence the reviewer is claiming. Note that if $A_t$ is constant, as for instance in [3], then this is indeed true, but crucially is not the case for the input-dependent ``selective'' setting of all recently proposed efficient RNNs such as Mamba.
>
> To detail more on expressivity in this context: it is easy to prove, see e.g. [4], that linear dense or diagonal RNNs with constant gates can only approximate linear functionals (path-to-point maps), even as the width increases to infinity. Instead, [2] shows theoretically that dense linear RNNs with input-dependent dense (i.e. non-constant and non-simultaneously-diagonalizable) gates can approximate arbitrary non-linear path-to-point maps as the width increases. Crucially, [2] also shows that the diagonal setting is distinct: one can approximate only specific input features that do not capture global non-linear interactions. [1] goes one step further and provides a concrete example task: state-tracking, which is not solvable with selective linear diagonal RNNs but can be solved with dense variants (see theory theorem 4.4).
>
> Overall, the studies discussed above, together with the classical results on universal approximation theorems, motivate our characterization of block-diagonal linear time-variant RNNs as forming a *hierarchy* of expressive power. We hope that this answers the reviewer's questions and are happy to discuss this further should this not be the case.
>
> **3. \& 4. Questions on optimal choice of $m$.**
> We thank the reviewer for raising the important question regarding the optimal choice of $m$ and we acknowledge that our initial submission did not sufficiently address this aspect. However, including this discussion can significantly enhance the practical value of our work. Accordingly, we have added an extended discussion on this topic to the Conclusion section (Appendix A).
>
> **4. On parameterization of models.**
> Our architecture uses and benefits from exponential reparameterization (softmax), one can find the exact implementation of our reparameterization in Appendix I. Addressing feedback from multiple reviewers, we have updated Figure 1 to enhance clarity. The figure now explicitly includes results for diagonal LRUs (BD-LRU and H-LRU with $m=1$) alongside Mamba2 [5,6], the state-of-the-art diagonal baseline that improves upon earlier SSMs like S4.
>
> **5. On the limitations of H-LRU.** In our discussion, we explicitly acknowledge the limitations of H-LRU, noting that its primary advantage lies in parameter efficiency rather than  peak performance. Additionally, Appendix G highlights the potential utility of non-selective H-LRU variants in heavily parameter-constrained settings, such as efficient embedding layers.
>
> References:
>
> [1] Merrill, W., Petty, J., \& Sabharwal, A. (2024). The illusion of state in state-space models. arXiv preprint arXiv:2404.08819.
>
> [2] Muca Cirone, N., Orvieto, A., Walker, B., Salvi, C., \& Lyons, T. (2024). Theoretical foundations of deep selective state-space models. Advances in Neural Information Processing Systems, 37, 127226-127272.
>
> [3] Orvieto, A., Smith, S. L., Gu, A., Fernando, A., Gulcehre, C., Pascanu, R., \& De, S. (2023, July). Resurrecting recurrent neural networks for long sequences. In International Conference on Machine Learning (pp. 26670-26698). PMLR.
>
> [4] Li, Z., Han, J., \& Li, Q. (2022). Approximation and optimization theory for linear continuous-time recurrent neural networks. Journal of Machine Learning Research, 23(42), 1-85.
>
> [5] Gu, A., \& Dao, T. (2023). Mamba: Linear-Time Sequence Modeling with Selective State Spaces. arXiv preprint arXiv:2312.00752.
>
> [6] Dao, T., \& Gu, A. (2024). Transformers are ssms: Generalized models and efficient algorithms through structured state space duality. arXiv preprint arXiv:2405.21060.

---

> > ### Comment · Reviewer_nE3j · 2025-11-28
> >
> > Thank the author for addressing my concerns.
> >
> > 1. For the questions about expressivity, I agree with the author's reply. Considering the time-variant setting for $a_t$, I think the results are indeed worth investigating. It is not as obvious as the time-invariant cases, where $a_t$ is constant through $t$.
> >
> > 2. My concerns about reparameterization are resolved.
> >
> >
> > 3. My major concern was about the H-LRU. Intuitively, this kind of method would loss certain memory properties and result in performance degrade. However, the author claim that this model could benifit in parameter constrained cases. I agree with this argument. However, to support this argument I think at least some toy example is needed. For example, showing in the parameter constrained setting, the H-LRU have advantage over regular models.
> >
> >
> > 4. For appendix G,  how is the kernel size and window size matched when comparing convolution and H-LRU. This is important for a reasonable comparision between this two method. Since if convolution got enough parameter, it should represent the H-LRU's constant kernel? (correct me if I got this wrong)
> >
> > Since my concerns are partially resolved, I would increase my score to 4.
> >
> > If the authors could explain 3 and 4 above, I will further increase my score.

---

> ### Author Response · Authors · 2025-12-02
>
> The authors thank the reviewer for acknowledging our efforts and increasing their score. In the following, we address the 2 remaining questions and hope that these revisions further improve the overall quality of our submission.
>
> Regarding point 3.:
>
> We now substantiate our claim of the parameter efficiency of the H-LRU model in the section "H-LRUs are parameter efficient" that refers to the results which are shown in Figure 5 (Appendix). These results demonstrate that H-LRU scales more efficiently than the LSTM baseline, which itself scales better than all other linear models tested, substantiating our claim.
> Specifically, in the parameter-constrained regime (400k parameters) H-LRU achieves a performance that surpasses all other baselines in the compression task (Fig. 5A). This is in contrast to BD-LRU which achieves a higher absolute peak performance but requires a significantly larger parameter budget to do so.
>
> To further validate parameter efficiency of H-LRU, we conducted additional language modeling experiments with using H-LRU and BD-LRU with matching parameter counts. Notably, in this fixed-parameter regime, H-LRU outperforms BD-LRU, confirming its superior parameter efficiency, although at a higher computational and memory cost. For a discussion and a presentation of the results see Appendix B.
>
> We can confirm the reviewer's prediction that H-LRU performance degrades with excessively large values of $m$, as detailed in Table 3 (Appendix) and Figure 1. We agree that extending the memory horizon beyond a certain point yields diminishing returns. This aligns with the central finding of our study: moderate window/block sizes tend to offer the optimal inductive bias for sequence modeling (and the optimal choice of $m$ is data set dependent).
>
> Regarding point 4.:
>
> We agree with the reviewer that a fair comparison based on parameter counts is crucial. To address this, we have introduced Figure 6B, which plots the performance of the models from Figure 6A against their parameter counts. A key observation here is a pronounced parameter efficiency of the non-selective H-LRU: scaling the window size improves performance with negligible parameter increase, resulting in a nearly vertical trajectory on the plot.
>
> Regarding the comparison to convolutions, the reviewer is correct that a time-invariant H-LRU can be conceptually viewed as a convolution with a kernel size equal to the sequence length. However, a critical distinction lies in parameter scaling. Unlike a standard convolution, which requires increasing the kernel parameters to match the sequence length, H-LRU adapts to any sequence length with a fixed number of parameters. Furthermore, the H-LRU weights are normalized by design, providing a distinct inductive bias compared to standard convolutional kernels. Thus, while H-LRU approximates a global convolution, its advantage lies in its ability to do so with significantly higher parameter efficiency. However, we also want to emphasize that non-selective models fail to match the performance of the selective architectures, which constitutes the primary focus of our study.
>
> We hope that these revisions allowed us to further improve the quality of our work and address the additional questions raised by the reviewer.

---

### Official Review · Reviewer_ZsLY · 2025-11-03

**Soundness:** 3
**Presentation:** 3
**Contribution:** 3
**Rating:** 8
**Confidence:** 3

**Summary:**

The paper studies linear RNNs with a more expressive transition than the common diagonal form. Higher-order LRUs (H-LRU) mix the last m hidden states, and block-diagonal LRUs (BD-LRU) use a dense transition matrix with m×m blocks.

They introduce a normalisation procedure for well-behaved training when scaling m, avoiding exploding/vanishing gradients.

On several synthetic sequence tasks, increasing m improves accuracy, especially for BD-LRUs. The biggest gains often appear when moving from m=1 to m=2. The authors attribute this to access to complex eigenvalues only when m>1.

Importantly, they provide an efficient higher-order parallel scan implementation for these non-diagonal LRUs.

**Strengths:**

The motivation is clear. Diagonal LRNNs are efficient but expressively limited. Structured non-diagonal mixing can close the gap while retaining much of the efficiency.

The normalisation is well motivated and supported by the ablation in Figure 2.

Several benchmarks support the move to H-LRUs and BD-LRUs. The jump from m=1 to m=2 is notable, plausibly due to complex eigenvalues (as the authors note).

Permutation tasks show an advantage for higher m as task complexity increases, especially for BD-LRU.

**Weaknesses:**

The use of H-LRU and BD-LRU themselves is not novel, which is why the efficient implementation and normalization are so important.

The main text briefly states how block-diagonal structure reduces the cost of the parallel scan; more detail is deferred to the appendix/code. A short sketch in the main text would help.

**Questions:**

If I understand correctly, BD-LRU and H-LRU are equivalent at m=1. In Figure 1 it would be good to highlight the m=1 case separately. Are the m=1 dots overlapping for H- and BD-LRU?

It might be too much to include full derivations in the main text, but since it is central to the study, could you briefly flesh out how the block-diagonal structure reduces the parallel-scan time complexity (the “hopscan” approach) in the main text, and point to the detailed appendix/code?

I would be curious to see the performance of across m for a benchmarks of real language data, such as wikitext-103. Would it be possible to include something like this?

Optional curiosity: have you considered a diagonal transition with complex entries as a baseline?

---

> ### Author Response · Authors · 2025-11-26
>
> We thank the reviewer for their valuable feedback and for their overall positive judgment, considering our contributions well-motivated and supported by the results. In the following, we address the main questions raised by the reviewer.
>
> **1) Parallel scan implementation.** We thank the reviewer for their feedback on the Implementation section which indeed was a bit brief. We have revised the description of our parallel scan algorithm to improve clarity and described all important details that were missing before.
>
> In the new version of the Implementation section, we present an improved description of the recurrence relation that is computed by our algorithm. Importantly, we also add the description of the associative operator and the substitution that we use to reduce our recurrence relation to a prefix sum, which makes application of Blelloch scan possible. In the updated Implementation section, we also present our extended throughput tests and an updated version of Figure.
>
> **2) Figure 1.** We thank the reviewer for pointing out that our Figure 1 can benefit from additional clarity. The reviewer is correct that BD-LRU and H-LRU are equivalent for $m=1$. Indeed, there are no substantial performance differences between H-LRU m1 and BD-LRU m1, as can be seen in Table 1. We attribute the small differences to differences in random initializations of the H-LRU and BD-LRU models.
>
> In an updated Figure 1, we now show a ``diagonal'' LRU which presents the best results across both H-LRU m1 and BD-LRU m1. We thank the reviewer for their helpful suggestions about Figure 1 and hope that the new version is able to increase the clarity of the presentation of our results.
>
> **3) Language modeling.** We agree with the reviewer that including a standard language modeling benchmark would strengthen our contribution. Therefore, we have added a new section (Appendix B) that presents language modeling experiments on the FineWeb dataset. These results further support the main findings of our work. If we have enough time and compute, we would like to run models with complex state transition matrices as additional baselines. We thank the reviewer for this interesting suggestion.

---

> > ### Author Response · Authors · 2025-12-02
> >
> > We hope that our prior response and the updated revision have thoroughly addressed the concerns raised. We remain open to any further discussion should additional points require clarification.

---

### Official Review · Reviewer_A46F · 2025-11-03

**Soundness:** 2
**Presentation:** 4
**Contribution:** 3
**Rating:** 4
**Confidence:** 3

**Summary:**

This paper introduces two Linear RNN gating parameterizations that enhance expressivity while retaining efficiency. The first one, Higher-order Linear Recurrent Units (H-LRU), generalizes first-order recurrence to m-th order, enabling temporal mixing across multiple past hidden states. The second one, Block-Diagonal Linear Recurrent Units (BD-LRU), replaces purely diagonal state-transition matrices with a block-diagonal structure, permitting richer intra-block channel mixing. Both architectures incorporate selective gating and L1-normalization of the transition blocks (per-channel for H-LRU, per-row for BD-LRU) to ensure stability. The authors also provide a theoretical guarantee (Proposition 1) that under this L1-normalized gating, the hidden-state norm remains bounded.

Empirical results show that both H-LRU and BD-LRU achieve strong performance on the MAD benchmark tasks (compression, associative recall and compression). Moreover, BD-LRU matches or exceeds recent linear baselines (Mamba, DeltaNet, DeltaProduct) and even LSTM on the permutation tasks that require state tracking. Finally, the authors provide a parallel-scan implementation that is shown to yield throughput comparable to diagonal LRNNs for moderate block sizes.

**Strengths:**

1. The proposed architectural parameterizations (H-LRU, BD-LRU) are conceptually elegant and clearly motivated. The authors show that higher-order recurrence and block structure are natural ways to increase mixing without resorting to full dense transitions. Both architectures incorporate input-dependent selective gates with L1-normalization (per-channel or per-row), ensuring forward-pass stability and bounded dynamics.

2. The authors provide a theoretical justification (Proposition 1) that this normalization guarantees dynamical stability and normalized hidden-state evolution, showcasing that these architectures are well-behaved and stable by construction.

3. Overall, the paper is well-organized. The progression from model definition (Sec 2) to normalization (Sec 3) and experiments (Sec 4, 5) is nicely structured. Figures and tables presenting results are clear too.

4. The results are strong on the considered synthetic tasks. BD-LRU, in particular, shows significant gains over existing linear RNNs (Mamba, DeltaNet and DeltaProduct) and LSTMs also on state-tracking tasks like permutation tasks (Table 2).

5. The proposed parallel-scan algorithm demonstrates that block-diagonal recurrences can achieve throughput competitive with diagonal ones (Figure 4), addressing a practical concern for deployment.

**Weaknesses:**

1. **Limited evaluations**: The tasks considered in the empirical evaluations are only synthetic ones. While the use of the MAD benchmark is very useful, demonstrating competitive performance on large-scale language or long-sequence datasets (e.g., LRA, language modeling). This makes it hard to assess if the structural benefits of the proposed architectures transfer to more practical usage. The authors acknowledge this as future work though.

2. The core contributions are about the structure of state mixing. Yet, the authors do not provide empirical analysis on the actual learned matrices and their properties, e.g., their eigenvalue spectra. This would provide crucial insight into what kind of mixing the model is learning (e.g., is BD-LRU learning negative eigenvalues?) and why it succeeds on state-tracking (permutation task).

3. The discussion mentions FLOPs and throughput but lacks detailed runtime comparisons across block sizes, hidden dims and sequence lengths. It’s unclear how block‐size m scales in practice (for m large).

4. The paper provides key code snippets in Appendix G, however, the full code is not yet released.

**Questions:**

1. To what extent can the BD-LRU architecture support eigenvalues outside the positive range (i.e., negative or complex eigenvalues) in the state transition matrix, as discussed by Grazzi et al. 2025? How does the L1 norm, block size m or order in H-LRU affect the eigenspectrum?

2. Outside the permutation tasks, have you also evaluated on regular language tasks such as parity or modular counting, where such eigenvalues were shown to be necessary for state-tracking?

3. Can you also provide latency/throughput for varying block sizes (e.g., m=2, 4, 8, 16) and higher sequence lengths to better assess the runtime trade-off for the parallel-scan implementation?


-- References --

*Gu, A., Dao, T. Mamba: Linear-time sequence modeling with selective state spaces. arXiv 2023*

*Orvieto, A., Smith, S.L., Gu, A., Fernando, A., Gulcehre, C., Pascanu, R., De, S.. Resurrecting recurrent neural networks for long sequences, in: International Conference on Machine Learning 2023*

*Gu, A., Goel, K., Ré, C.. Efficiently modeling long sequences with structured state spaces. arXiv 2021*

*J. Siems, T. Carstensen, A. Zela, F. Hutter, M. Pontil, R. Grazzi. DeltaProduct: Improving State-Tracking in Linear RNNs via Householder Products. In NeurIPS 2025.*

*R. Grazzi, J. Siems, A. Zela, J.K.Franke, F. Hutter, M. Pontil. Unlocking State-Tracking in Linear RNNs Through Negative Eigenvalues. In ICLR 2025*

*M. Poli, A. W. Thomas, E. Nguyen, et al. Mechanistic Design and Scaling of Hybrid Architectures. arXiv 2024*

---

> ### Author Response · Authors · 2025-11-26
>
> We thank the reviewer for their valuable feedback and for considering our contributions conceptually elegant and clearly motivated. In the following, we address the main questions raised by the reviewer.
>
> **1) Eigenvalues spectra.** We thank the reviewer for their helpful suggestions and comments on the properties of the eigenvalue spectra of the transition matrices. To address the comment, we have performed additional analyses on the eigenvalues of the transition matrices and have added our findings to Appendix H.
>
> The eigenvalue analysis revealed that the proposed H-/BD-LRU architectures gain access to negative eigenvalues starting from $m=2$ and to complex eigenvalues starting from $m=3$. The improvement in performance between $m=1$ and $m=2$ aligns well with previous results obtained by Grazzi [1]. We have updated the main text, discussing the new findings.
>
> We thank the reviewer for raising this important point that we think adds additional value and brings about useful insights.
>
> **2) Language modeling.** We agree with the reviewer that including a standard language modeling benchmark would strengthen our contribution. Therefore, we have added a new section presenting language modeling experiments on the FineWeb dataset. These results further support the main findings of our work and now discussed in Appendix A (Conclusion and Discussion) and Appendix B (Language Modeling).
>
> **3) Extended throughput tests.** We thank the reviewer for their suggestion to extend our throughput tests. We present the results of our extended tests in the Implementation part and in the updated Figure 3. We have also revised the description of our parallel scan algorithm to increase clarity.
>
> The new results in Figure 3B present a comparison between models with the same hidden dimension 768 (base size of GPT2) across different block sizes (1, 2, 4, 8, 16, 64, 128, 256). They reveal that throughput degradation is negligible if the window size is less than 16, and increases for larger window sizes. Meanwhile, Figure 3C confirms that at the $\sim$33M parameter scale (one layer of similar size as in a 1B GPT2 model), models with small block sizes (2–5) achieve lower throughput than Mamba, likely due to hardware utilization constraints rather than theoretical complexity, as discussed in the revised text.
>
> **4) Regular formal language tasks.** We thank the reviewer for proposing to consider regular language tasks such as parity or modular counting. These tasks are also part of Chomsky Hierarchy tasks as described in [3]. We have added a presentation of the results that we obtained on Parity, Modular Arithmetic with and without brackets and Cycle Navigation tasks in Appendix K. Our experiments on these tasks are in good agreement with our previous results on the permutation tasks.
>
>  The results on the Parity task and the Cycle Navigation task provide further support for the previously found advantage of BD-LRUs on permutations tasks. In our experiments on Modular Arithmetic tasks, we found that BD-LRU were able to achieve performance levels similar to the ones of the other best performing RNNs [2,3].
>
> References:
>
> [1] Grazzi, R., Siems, J., Zela, A., Franke, J. K., Hutter, F., \& Pontil, M. (2024). Unlocking state-tracking in linear rnns through negative eigenvalues. arXiv preprint arXiv:2411.12537.
>
> [2] Siems, J., Carstensen, T., Zela, A., Hutter, F., Pontil, M., \& Grazzi, R. (2025). Deltaproduct: Improving state-tracking in linear rnns via householder products. arXiv preprint arXiv:2502.10297.
>
> [3] Delétang, G., Ruoss, A., Grau-Moya, J., Genewein, T., Wenliang, L. K., Catt, E., ... \& Ortega, P. A. (2022). Neural networks and the chomsky hierarchy. arXiv preprint arXiv:2207.02098.

---

> > ### Author Response · Authors · 2025-12-02
> >
> > We hope that our prior response and the updated revision have thoroughly addressed the concerns raised. We remain open to any further discussion should additional points require clarification.

---

### Author Response · Authors · 2025-12-03
**General Reply to the Reviewers**

We thank the reviewers for their constructive feedback. A recurring theme in the reviews was the need to validate the performance of our proposed H-LRUs and BD-LRUs on language modeling, and to deepen the analysis of their computational efficiency. We have addressed these concerns comprehensively. We have extended the manuscript with new sets of experiments and have significantly expanded the comparisons between the different models and model variants. Specifically we have added:

**1) Language Modeling (Reviewers A46F, ZsLY, E3FU):** We have moved beyond synthetic benchmarks and added a new section (Appendix B) presenting results on the FineWeb dataset. These experiments confirm that the structural benefits of BD-LRU and H-LRU transfer to real data sets such as language modeling.

**2) Implementation \& Throughput (Reviewers A46F, ZsLY, E3FU):** We rewrote the Implementation section to provide a detailed description of the parallel scan reduction used. We also updated Figure 3 with extended throughput tests, covering a wider range of block sizes ($m \in [1, 256]$) and providing comparison with parameter-matched layers ($\sim 33M$).

**3) Hidden sizes and block sizes (Reviewers A46F, nE3j, E3FU):** We have expanded our experiments for both synthetic and language modeling tasks to include larger hidden dimensions (up to 6k) and extended block sizes (up to $m=16$). Figure 1 and Tables 1,2,3,4 have been updated to include these new findings. We also have added an extended discussion on the optimal choice of $m$ to the Conclusion section.

**4) Extended Formal Languages (Reviewer A46F):** We have expanded the evaluation to include Regular Language tasks (Parity, Modular Arithmetic, Cycle Navigation) in Appendix K, demonstrating how our architectures meet the requirements of the Chomsky hierarchy.

**5) Spectral Analysis (Reviewer A46F):** We have added an empirical analysis of the learned transition matrices (Appendix H). We now show that H-LRU and BD-LRU gain access to negative eigenvalues starting at block size $m=2$ and complex eigenvalues at $m=3$, explaining the performance jumps observed in state-tracking tasks.

**6) Normalization (Reviewers nE3j, E3FU):** We have addressed the reviewers' questions regarding reparameterization and normalization. We have improved the stability of the ReLU normalization implementation. These updated results have been incorporated into Figure 2 and the main text in the Normalization section.

**7) Presentation of results on Figure 1 (Reviewers A46F, ZsLY, nE3j, E3FU):** We have significantly revised Figure 1 to address multiple reviewer comments. To enhance clarity, the figure now explicitly contrasts results for diagonal LRUs against the best-performing BD-LRU and H-LRU configurations. To Figure 1, we have also added plots that compare different block and window sizes separately.

**8) Parameter efficiency and expressivity (Reviewer nE3j):** We have introduced new language modeling experiments in Appendix B to further substantiate the parameter efficiency of H-LRU. Additionally, we have included a new plot in the ablation study to highlight the parameter efficiency of non-selective models (Figure 6B, Appendix). We have also addressed the reviewers comments on the expressivity of selective and non-selective models in our comments.

---

### Meta-Review · Area_Chair_dMmL · 2026-01-06

**Summary:**

Strengths:
1. Clear motivation of improving expressivity of LRNNs
2. The proposed normalization scheme stabilizes training
3. Strong performance on synthetic benchmarks

Main concerns:
1. (A46F, ZsLY, E3FU) Limited evaluation on synehtic benchmarks only
2. (A46F) Lack of empirical analysis on the actual learned matrices,
3. (A46F, nE3j, E3FU) Lack of ablation and design choice of hyperparameter m
4. (nE3j) Confusion of additional expressivity with dense matrices
5. (nE3j) Inferior performance of H-LRU
6. (E3FU) Dense writing style
7. (E3FU) Confusion of extremely small scale in the block size.
8. (A46F, ZsLY, nE3j, E3FU) Confusion of the performance of BD-LRU in Figure 1
9. (A46F, ZsLY, E3FU) Implementation of parallel scan and throughput analysis

**Reviewer Concerns:**

Concerns 2, 3, 4, 7, 8 and 9 should have been addressed properly after rebuttal.

Concern 1 is the most critical weakness of this submission. Although the authors provided a real world language modeling experiment on FineWeb dataset, the experiment results are still limited, restricted to small model sizes and lack of comparison to standard baselines. The additional section in the appendix is not enough to clear the shared concern from most reviewers. This section should be expanded with more results and moved to the main text.

Concern 5, lack inferior performance of H-LRU is another main concern. The authors proposed two architetures, H-LRU and BD-LRU. While BD-LRU shows strong performance with a proper choice of block size, the performance of H-LRU is generally worse than baselines. While the authors argue H-LRU has more parameter efficiency, it's only demonstrated on the compression task (Figure 5A) which is hard to assess its significance in practice.

Concern 6 on the writing style is not properly discussed in the rebuttal but it's not a main concern for the decision of the submission.

**Reviewer Scores:**

A46F, 4. prediction: 5
The majority of concerns have been addressed, but the main concern of real world evaluated is not resolved.

ZsLY, 8. Prediction: 8
The reviewer is already supportive of this submission.

nE3j, initial 2, raised to 4 during discussion, predictoin: 4. The remaining concern about the performance of H-LRU is not fully addressed.

E3FU, 2. Prediction: 4
The misunderstanding on the "extremely small scale" of the block size has been clarified by the rebuttal. The concerns on the reliance on synthetic problems and writing style are not addressed satisfactorily.

---

### Decision · Program_Chairs · 2026-01-26

Reject